# Assessing Robustness via Score-Based Adversarial Image Generation

**Marcel Kollovieh**[1,2,3], **Lukas Gosch**[1,2,3], **Marten Lienen**[1], **Yan Scholten**[1,2], **Leo Schwinn**[1,2],
**Stephan Günnemann**[1,2,3]                                                    *m.kollovieh@tum.de*
[1]*School of Computation, Information and Technology, Technical University of Munich,* [2]*Munich Data Science Institute,* [3]*Munich Center for Machine Learning*

**Reviewed on OpenReview:** *https://openreview.net/forum?id=7Oqb6zlGWl*

## Abstract

Most adversarial attacks and defenses focus on perturbations within small $\ell_p$-norm constraints. However, $\ell_p$ threat models cannot capture all relevant semantics-preserving perturbations, and hence, the scope of robustness evaluations is limited. In this work, we introduce Score-Based Adversarial Generation (ScoreAG), a novel framework that leverages the advancements in score-based generative models to generate unrestricted adversarial examples that overcome the limitations of $\ell_p$-norm constraints. Unlike traditional methods, ScoreAG maintains the core semantics of images while generating adversarial examples, either by transforming existing images or synthesizing new ones entirely from scratch. We further exploit the generative capability of ScoreAG to purify images, empirically enhancing the robustness of classifiers. Our extensive empirical evaluation demonstrates that ScoreAG improves upon the majority of state-of-the-art attacks and defenses across multiple benchmarks. This work highlights the importance of investigating adversarial examples bounded by semantics rather than $\ell_p$-norm constraints. ScoreAG represents an important step towards more encompassing robustness assessments.

## 1 Introduction

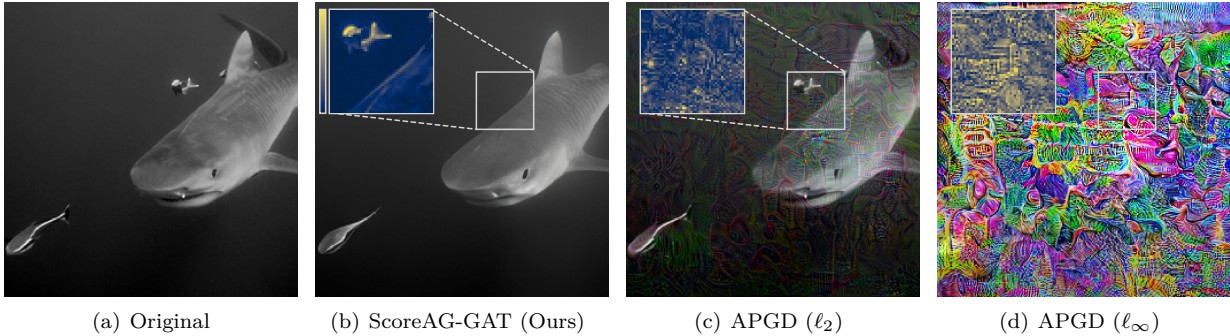

|  |  |  |  |
|---|---|---|---|
| (a) Original | (b) ScoreAG-GAT (Ours) | (c) APGD ($\ell_2$) | (d) APGD ($\ell_\infty$) |

Figure 1: Examples of various adversarial attacks on an image of the class "tiger shark" (a). The inset visualizes a heatmap of the strength of the corresponding perturbation. Despite the fact that the perturbation generated by ScoreAG-GAT (b) lies outside of common $\ell_p$-norm constraints ($\ell_\infty = 188/255$, $\ell_2 = 18.47$), it is aware of the semantics: removing a small fish to change the predicted label to "hammer shark". This is in stark contrast to APGD (Croce & Hein, 2020b) with matching norm constraints, which either (c) results in highly perceptible and unnatural changes, or (d) fails to preserve image semantics completely. This is an example of Generative Adversarial Transformation (GAS), one of the three use-cases of ScoreAG.

Ensuring robustness against noisy data or malicious interventions has become a major concern in various applications ranging from autonomous driving (Eykholt et al., 2018) and medical diagnostics (Dong et al., 2023) to the financial sector (Fursov et al., 2021). Even though adversarial robustness has received significant research attention (Goodfellow et al., 2014; Madry et al., 2017; Croce & Hein, 2020b), it is still an unsolved problem. Most works on adversarial robustness define adversarial perturbations to lie within a restricted $\ell_p$-norm from the input. However, recent works have shown that significant semantic changes can occur within common perturbation norms, and that many relevant semantics-preserving corruptions lie outside specific norm ball choices (Tramèr et al., 2020; Gosch et al., 2023). Examples include physical perturbations such as stickers on stop signs (Eykholt et al., 2018) or natural corruptions such as lighting or fog (Kar et al., 2022; Hendrycks & Dietterich, 2019). This led to the inclusion of a first $\ell_p$-norm independent robustness benchmark to RobustBench (Croce et al., 2020), and a call to further investigation into robustness beyond $\ell_p$-bounded adversaries (Hendrycks et al., 2022). Thus, in this work, we address the following research question:

*How can we generate semantics-preserving adversarial examples beyond $\ell_p$-norm constraints?*

We propose to leverage the significant progress in diffusion models (Sohl-Dickstein et al., 2015; Ho et al., 2020) and score-based generative models (Song et al., 2020) in generating realistic images. Specifically, we introduce *Score-Based Adversarial Generation (ScoreAG)*, a framework designed to synthesize adversarial examples, transform existing images into adversarial ones, and purify images. Using diffusion guidance (Dhariwal & Nichol, 2021), ScoreAG can generate semantics-preserving adversarial examples that are not captured by common $\ell_p$-norms (see Fig. 1). Overall, ScoreAG represents a novel tool for assessing and enhancing the empirical robustness of image classifiers.

Our *key contributions* are summarized as follows:

- We overcome limitations of $\ell_p$ threat models by proposing ScoreAG, a framework utilizing diffusion guidance on pre-trained models, enabling the generation of *unrestricted* but semantics-preserving adversarial examples.

- With ScoreAG we *transform* existing images into adversarial ones as well as *synthesize* completely new adversarial examples.

- We show that ScoreAG enhances classifier robustness by *purifying* adversarial examples and common corruptions.

- We demonstrate ScoreAG's capability in an exhaustive empirical evaluation and show it is able to outperform a majority of existing attacks and defenses on several benchmarks. Additionally, we underscore ScoreAG's semantic preserving ability in a human study.

## 2 Background

**Score-Based Generative Modelling.** Score-based generative models (Song et al., 2020) are a class of generative models based on a continuous-time diffusion process $\{\mathbf{x}_t\}_{t \in [0,1]}$ accompanied by their corresponding probability densities $p_t(\mathbf{x})$. The diffusion process progressively perturbs a data distribution $\mathbf{x}_0 \sim p_0$ into a prior distribution $\mathbf{x}_1 \sim p_1$. This transformation is formalized as a Stochastic Differential Equation (SDE), i.e.,

$$\mathrm{d}\mathbf{x}_t = \mathbf{f}(\mathbf{x}_t, t)\mathrm{d}t + g(t)\mathrm{d}\mathbf{w}, \tag{1}$$

where $\mathbf{f}(\cdot, t) : \mathbb{R}^d \to \mathbb{R}^d$ represents the drift coefficient of $\mathbf{x}_t$, $g(\cdot) : \mathbb{R} \to \mathbb{R}$ the diffusion coefficient, and $\mathbf{w}$ the standard Wiener process (i.e., Brownian motion). Furthermore, let $p_{st}(\mathbf{x}_t \mid \mathbf{x}_s)$ describe the transition kernel from $\mathbf{x}_s$ to $\mathbf{x}_t$, where $s < t$.

By appropriately choosing $\mathbf{f}$ and $g$, $p_1$ asymptotically converges to an isotropic Gaussian distribution, i.e., $p_1 \approx \mathcal{N}(\mathbf{0}, \mathbf{I})$. To generate data, the reverse-time SDE needs to be solved:

$$\mathrm{d}\mathbf{x}_t = [\mathbf{f}(\mathbf{x}_t, t) - g(t)^2 \nabla_{\mathbf{x}_t} \log p_t(\mathbf{x}_t)]\mathrm{d}t + g(t)\mathrm{d}\mathbf{w}. \tag{2}$$

Solving the SDE requires access to the time-dependent score function $\nabla_{\mathbf{x}_t} \log p_t(\mathbf{x}_t)$, which is typically unknown. Instead, the score function is estimated using a neural network $\mathbf{s}_\theta(\mathbf{x}_t, t)$. The parameters of this network are learned by minimizing the following cost function:

$$\mathbb{E}_t\left[\lambda(t)\mathbb{E}_{\mathbf{x}_0}\mathbb{E}_{\mathbf{x}_t|\mathbf{x}_0}\left[\|\mathbf{s}_\theta(\mathbf{x}_t, t) - \nabla_{\mathbf{x}_t} \log p_{0t}(\mathbf{x}_t \mid \mathbf{x}_0)\|_2^2\right]\right]. \tag{3}$$

Here, $\lambda(\cdot) : [0, 1] \to \mathbb{R}_{>0}$ serves as a time-dependent weighting parameter, and $t$ is uniformly sampled from the interval $[0, 1]$.

In this formulation, $\mathbf{x}_0 \sim p_0$ is sampled from the data distribution, and $\mathbf{x}_t \sim p_{0t}(\mathbf{x}_t \mid \mathbf{x}_0)$ follows the diffusion process at time $t$. The goal is to train the network $\mathbf{s}_\theta$ to accurately match the true score function $\nabla_{\mathbf{x}_t} \log p_{0t}(\mathbf{x}_t \mid \mathbf{x}_0)$, enabling data generation through the reverse diffusion process, which can be solved using numerical solvers.

**Diffusion Guidance.** To enable conditional generation with unconditionally trained diffusion models, Dhariwal & Nichol (2021) introduce classifier guidance. The central idea is to generate samples from the conditional distribution $p(\mathbf{x}_0 \mid c)$, where $c$ represents a specific class, i.e., sampling images of class $c$. To achieve this, the authors replace the gradient of the unconditional distribution $p_t(\mathbf{x}_t)$ in the reverse process (see equation 2) with its conditional counterpart.

By applying Bayes' theorem, the gradient of the conditional gradient can be decomposed as:

$$\nabla_{\mathbf{x}_t} \log p(\mathbf{x}_t \mid c) = \nabla_{\mathbf{x}_t} \log p(\mathbf{x}_t) + \nabla_{\mathbf{x}_t} \log p(c \mid \mathbf{x}_t), \tag{4}$$

where $\nabla_{\mathbf{x}_t} \log p(\mathbf{x}_t)$ represents unconditional score function and $\nabla_{\mathbf{x}_t} \log p(c \mid \mathbf{x}_t)$ represents the guidance score. The unconditional score function is approximated using the neural network $\mathbf{s}_\theta$, which is trained using the loss in equation 3.

To compute the guidance score $\nabla_{\mathbf{x}_t} \log p(c \mid \mathbf{x}_t)$, Dhariwal & Nichol (2021) utilize the gradients of a time-dependent classifier $f(\mathbf{x}_t, t)$ with respect to $\mathbf{x}_t$. The guidance score steers the generation process towards samples that are consistent with the desired class $c$. This method allows an unconditional diffusion model, i.e., a model trained without conditional information, to be adapted for conditional tasks, enabling the generation of class-specific samples.

Classifier guidance has since been extended to handle arbitrary conditions $c$, such as guiding generation towards CLIP embeddings (Nichol et al., 2021). This flexibility in choosing different conditions is essential to ScoreAG and enables us to adapt the model for three distinct tasks by adjusting the guidance condition, as described in the next section.

## 3 Score-Based Adversarial Generation

In this section, we introduce *Score-Based Adversarial Generation* (ScoreAG), a framework employing generative models to evaluate robustness beyond the $\ell_p$-norm constraints. ScoreAG is designed to perform the following three tasks: **(1)** the generation of adversarial images (see Sec. 3.2), **(2)** the transformation of existing images into adversarial examples (see Sec. 3.3), and **(3)** the purification of images to enhance empirical robustness of classifiers (see Sec. 3.4).

ScoreAG consists of three steps: **(1)** select a guidance term for the corresponding task to model the conditional score function $\nabla_{\mathbf{x}_t} \log p(\mathbf{x}_t \mid c)$, **(2)** adapt the reverse-time SDE with the task-specific conditional score function, and **(3)** solve the adapted reverse-time SDE for an initial noisy image $\mathbf{x}_1 \sim \mathcal{N}(\mathbf{0}, \mathbf{I})$ using numerical methods. Depending on the task, the result is either an adversarial or a purified image. We provide an overview of ScoreAG in Fig. 2.

In detail, the conditional score function is composed of the normal score function $\nabla_{\mathbf{x}_t} \log p_t(\mathbf{x}_t)$ and the task-specific guidance term $\nabla_{\mathbf{x}_t} \log p(c \mid \mathbf{x}_t)$, that is

$$\nabla_{\mathbf{x}_t} \log p_t(\mathbf{x}_t \mid c) = \nabla_{\mathbf{x}_t} \log p_t(\mathbf{x}_t) + \nabla_{\mathbf{x}_t} \log p_t(c \mid \mathbf{x}_t), \tag{5}$$

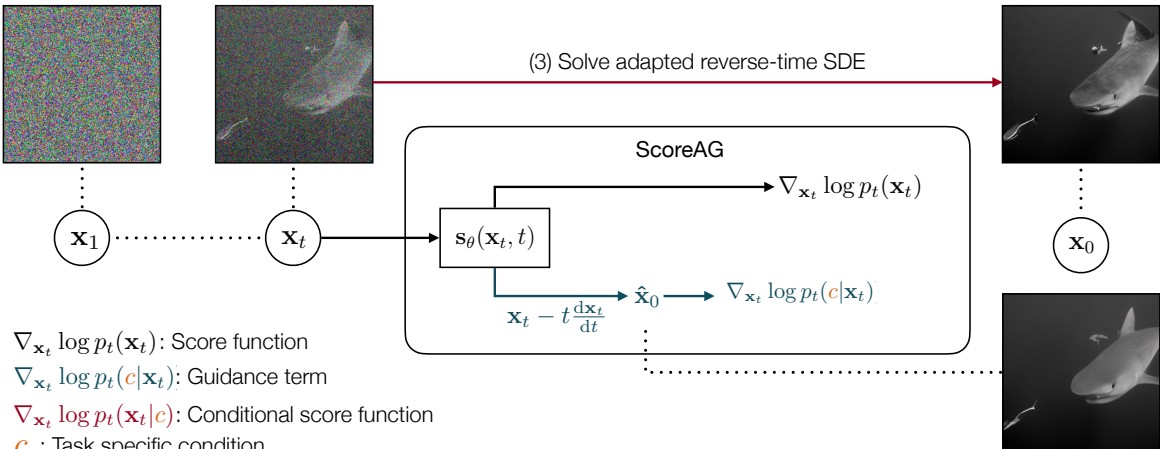

(1) Select task-specific guidance term $\nabla_{\mathbf{x}_t} \log p_t(c|\mathbf{x}_t)$ to model the conditional score function: $\nabla_{\mathbf{x}_t} \log p_t(\mathbf{x}_t|c)$

(2) Adapt reverse-time SDE $\quad \mathrm{d}\mathbf{x}_t = [\mathbf{f}(\mathbf{x}_t, t) - g(t)^2 \nabla_{\mathbf{x}_t} \log p_t(\mathbf{x}_t|c)]\mathrm{d}t + g(t)\mathrm{d}\mathbf{w}$

(3) Solve adapted reverse-time SDE

ScoreAG

$\nabla_{\mathbf{x}_t} \log p_t(\mathbf{x}_t)$

$\mathbf{s}_\theta(\mathbf{x}_t, t)$

$\mathbf{x}_1$    $\mathbf{x}_t$

$\mathbf{x}_t - t\frac{\mathrm{d}\mathbf{x}_t}{\mathrm{d}t}$   $\hat{\mathbf{x}}_0$   $\nabla_{\mathbf{x}_t} \log p_t(c|\mathbf{x}_t)$

$\mathbf{x}_0$

$\nabla_{\mathbf{x}_t} \log p_t(\mathbf{x}_t)$: Score function
$\nabla_{\mathbf{x}_t} \log p_t(c|\mathbf{x}_t)$: Guidance term
$\nabla_{\mathbf{x}_t} \log p_t(\mathbf{x}_t|c)$: Conditional score function
$c$ : Task specific condition

Figure 2: An overview of ScoreAG and its three steps. ScoreAG starts from noise $\mathbf{x}_1$ and iteratively denoises it into an image $\mathbf{x}_0$. It uses the task-specific guidance terms $\nabla_{\mathbf{x}_t} \log p_t(c \mid \mathbf{x}_t)$ and the score function $\nabla_{\mathbf{x}_t} \log p_t(\mathbf{x}_t)$ to guide the process towards the task specific condition $c$. The network $\mathbf{s}_\theta$ is used for approximating the score function $\nabla_{\mathbf{x}_t} \log p_t(\mathbf{x}_t)$ and for the one-step Euler prediction $\hat{\mathbf{x}}_0$.

where $\log p_t(\mathbf{x}_t)$ is modeled by a score-based generative model. Solving the adapted reverse-time SDE yields a sample of the conditional distribution $p(\mathbf{x}_0 \mid c)$, i.e., an adversarial or purified image. To simplify the presentation, we will denote class-conditional functions as $p_y(\mathbf{x}_t)$ rather than the more verbose $p(\mathbf{x}_t \mid y)$.

## 3.1 Problem Statement.

In the realm of adversarial robustness, traditional evaluation methods often constrain adversarial perturbations within an $\ell_p$-norm ball, providing a limited robustness assessment. These limitations are addressed by *unrestricted* attacks. In this work, we consider the following three key tasks: **(1)** Generating new adversarial images that inherently belong to a specific class $y^*$ but are misclassified by the classifier as $\tilde{y}$; **(2)** Transforming existing images $\mathbf{x}^*$ into adversarial examples, i.e., images that are misclassified as $\tilde{y}$ (see adversary) while maintaining their core semantics and true class $y^*$; and **(3)** Purifying adversarial images $\mathbf{x}_{\mathrm{ADV}}$ to recover correct classification and enhance empirical robustness.

**Adversary.** Let $y^* \in \{1, \ldots, K\}$ denote the true class of a clean image $\mathbf{x} \in [0, 1]^{C \times H \times W}$, $\tilde{y} \neq y^*$ be a different class, and $f(\cdot) : [0, 1]^{C \times H \times W} \to \{1, \ldots, K\}$ a classifier. An image $\mathbf{x}_{\mathrm{ADV}} \in [0, 1]^{C \times H \times W}$ is termed an adversarial example if it is misclassified by $f$, i.e., $f(\mathbf{x}) = y^* \neq \tilde{y} = f(\mathbf{x}_{\mathrm{ADV}})$, while preserving the semantics, i.e., $\Omega(\mathbf{x}) = \Omega(\mathbf{x}_{\mathrm{ADV}})$ with $\Omega$ denoting a semantics-describing oracle. Therefore, adversarial examples do not change the true label of the image. To enforce this, conventional adversarial attacks restrict the perturbation to lie in a certain $\ell_p$-norm, avoiding large differences to the original image. In contrast, ScoreAG is not limited by $\ell_p$-norm restrictions but preserves the semantics by employing a class-conditional generative model. In the following, we introduce each task in detail.

## 3.2 Generative Adversarial Synthesis

Generative Adversarial Synthesis (GAS) aims to synthesize images that are adversarial by nature. While these images maintain the semantics of a certain class $y^*$, they are misclassified by a classifier into a different class $\tilde{y}$. The formal objective of GAS is to sample from the distribution $p_{y^*}(\mathbf{x}_0 \mid f(\mathbf{x}_0) = \tilde{y})$, where $f(\mathbf{x}_0) = \tilde{y}$ corresponds to the guidance condition $c$.

Applying Bayes' theorem according to equation 5, the conditional score can be expressed as:

$$\nabla_{\mathbf{x}_t} \log p_{t,y^*}(\mathbf{x}_t \mid f(\mathbf{x}_0) = \tilde{y}) = \nabla_{\mathbf{x}_t} \log p_{t,y^*}(\mathbf{x}_t) + s_{\mathbf{y}} \cdot \nabla_{\mathbf{x}_t} \log p_{t,y^*}(f(\mathbf{x}_0) = \tilde{y} \mid \mathbf{x}_t), \tag{6}$$

where $s_{\mathbf{y}}$ is a scaling parameter adjusting the strength of the attack. While $\nabla_{\mathbf{x}} \log p_{t,y^*}(\mathbf{x}_t)$ can be learned with a class-conditional score network $\mathbf{s}_\theta(\mathbf{x}_t, t, y)$, $\nabla_{\mathbf{x}_t} \log p_{t,y^*}(f(\mathbf{x}_0) = \tilde{y} \mid \mathbf{x}_t)$ requires further analysis. By marginalizing over $\mathbf{x}_0$ and using the Markov property that $f(\mathbf{x}_0)$ and $\mathbf{x}_t$ are independent given $\mathbf{x}_0$, we see that

$$p_{t,y^*}(f(\mathbf{x}_0) = \tilde{y} \mid \mathbf{x}_t) = \mathbb{E}_{\mathbf{x}_0 \mid \mathbf{x}_t, t, y^*}[p(f(\mathbf{x}_0) = \tilde{y} \mid \mathbf{x}_0)] \tag{7}$$

is the expected probability of classifying generated samples $\mathbf{x}_0$ as class $\tilde{y}$.

While a direct Monte Carlo approximation to equation 7 is theoretically feasible, drawing samples from the class-conditional generative model $p_{t,y^*}(\mathbf{x}_0 \mid \mathbf{x}_t)$ would be expensive. Instead, we approximate $p_{t,y^*}(\mathbf{x}_0 \mid \mathbf{x}_t)$ as a Dirac distribution centered on the one-step Euler solution $\hat{\mathbf{x}}_0$ to equation 1 from $t$ to 0 $\hat{\mathbf{x}}_0 = \mathbf{x}_t - t\frac{\mathrm{d}\mathbf{x}_t}{\mathrm{d}t}$, which simplifies equation 7 to

$$p_{t,y^*}(f(\mathbf{x}_0) = \tilde{y} \mid \mathbf{x}_t) \approx p(f(\hat{\mathbf{x}}_0) = \tilde{y} \mid \mathbf{x}_t). \tag{8}$$

Thus, we approximate $\nabla_{\mathbf{x}_t} \log p_{t,y^*}(f(\mathbf{x}_0) = \tilde{y} \mid \mathbf{x}_t) \approx \nabla_{\mathbf{x}_t} \log p(f(\hat{\mathbf{x}}_0) = \tilde{y} \mid \mathbf{x}_t)$, which, in practice, corresponds to maximizing the cross-entropy between the classification $f(\hat{\mathbf{x}}_0)$ of the generated sample and the target class $\tilde{y}$.

In contrast to Dhariwal & Nichol (2021), our approximation allows us to work with the classifier $f$ directly instead of fine-tuning a time-dependent variant. Moreover, this can be adapted to discrete-time diffusion models with the approach by Kollovieh et al. (2023).

### 3.3 Generative Adversarial Transformation

While in GAS we synthesize adversarial samples from scratch, Generative Adversarial Transformation (GAT) focuses on transforming existing images into adversarial examples. For a given image $\mathbf{x}^*$ and its corresponding true class label $y^*$, the objective is to sample a perturbed image misclassified as $\tilde{y}$ while preserving the core semantics of $\mathbf{x}^*$. We denote the resulting distribution as $p_{y^*}(\mathbf{x}_0 \mid f(\mathbf{x}_0) = \tilde{y}, \mathbf{x}^*)$ for the guidance condition $c = \{\mathbf{x}^*, f(\mathbf{x}_0) = \tilde{y}\}$ leading to the following conditional score (equation 5):

$$\nabla_{\mathbf{x}_t} \log p_{t,y^*}(\mathbf{x}_t \mid \mathbf{x}^*, f(\mathbf{x}_0) = \tilde{y}) = \nabla_{\mathbf{x}_t} \log p_{t,y^*}(\mathbf{x}_t) + \nabla_{\mathbf{x}_t} \log p_{t,y^*}(\mathbf{x}^*, f(\mathbf{x}_0) = \tilde{y} \mid \mathbf{x}_t). \tag{9}$$

By assuming independence between $\mathbf{x}^*$ and $\tilde{y}$ given $\mathbf{x}_t$, we split the guidance term into $s_{\mathbf{x}} \cdot \nabla_{\mathbf{x}_t} \log p_{t,y^*}(\mathbf{x}^* \mid \mathbf{x}_t) + s_{\mathbf{y}} \cdot \nabla_{\mathbf{x}_t} \log p_{t,y^*}(f(\mathbf{x}_0) = \tilde{y} \mid \mathbf{x}_t)$, implying that $\tilde{y}$ should not influence the core semantics of the given image. Note that we introduced the two scaling parameters $s_{\mathbf{x}}$ and $s_{\mathbf{y}}$ to control the possible deviation from the original image and the strength of the attack, respectively. While we treat the score function $\nabla_{\mathbf{x}_t} \log p_{t,y^*}(\mathbf{x}_t)$ and the guidance term $\nabla_{\mathbf{x}_t} \log p_{t,y^*}(f(\mathbf{x}_0) = \tilde{y} \mid \mathbf{x}_t)$ as in the GAS setup, we model the distribution $p_{t,y^*}(\mathbf{x}^* \mid \mathbf{x}_t)$ as a Gaussian centered at the one-step Euler prediction $\hat{\mathbf{x}}_0$ (equation 8),

$$p_{t,y^*}(\mathbf{x}^* \mid \mathbf{x}_t) = \mathcal{N}(\hat{\mathbf{x}}_0, \mathbf{I}). \tag{10}$$

It follows that our sampling process searches for an adversarial example while minimizing the squared error between $\mathbf{x}^*$ and $\hat{\mathbf{x}}_0$. Importantly, this lets us generate samples $\mathbf{x}_0$ close to $\mathbf{x}^*$ without imposing specific $\ell_p$-norm constraints. Furthermore, our framework is not limited to the squared error, but can also utilize other differentiable similarity metrics as guidance such as the LPIPS (Zhang et al., 2018) score. Note that, while adversarial examples generated by ScoreAG are unrestricted in the sense of $\ell_p$-balls, they are constrained to the data manifold of the generative model through the construction of our generative process. This yields an *unrestricted* attack that preserves the core semantics using the class-conditional score network $\mathbf{s}_\theta$.

As a result, GAT provides a more comprehensive robustness assessment than traditional $\ell_p$-threat models. This enhanced assessment capability stems from the inherent properties of GAT, which (1) encompasses all semantics-preserving adversarial examples within the $\ell_p$-balls as captured by the generative model, and (2) includes semantics-preserving adversarial examples that conventional $\ell_p$-threat models do not capture.

### 3.4 Generative Adversarial Purification

Generative Adversarial Purification (GAP) extends ScoreAG to counter adversarial attacks. It is designed to purify adversarial images, i.e., remove adversarial perturbations through its generative capability to enhance the robustness of machine learning models.

Given an adversarial image $\mathbf{x}_{\text{ADV}}$ that was perturbed to induce a misclassification, GAP aims to sample an image from the data distribution that resembles the semantics of $\mathbf{x}_{\text{ADV}}$, which we denote as $p(\mathbf{x}_0 \mid \mathbf{x}_{\text{ADV}})$ with $\mathbf{x}_{\text{ADV}}$ corresponding to the guidance condition $c$. We model its score function analogously to equation 9:

$$\nabla_{\mathbf{x}_t} \log p_t(\mathbf{x}_t \mid \mathbf{x}_{\text{ADV}}) = \nabla_{\mathbf{x}_t} \log p_t(\mathbf{x}_t) + s_{\mathbf{x}} \cdot \nabla_{\mathbf{x}_t} \log p_t(\mathbf{x}_{\text{ADV}} \mid \mathbf{x}_t), \tag{11}$$

where $s_{\mathbf{x}}$ is a scaling parameter controlling the deviation from the input. Note that we omit $y^*$ since there is no known ground-truth class label. As previously, we utilize a time-dependent score network $\mathbf{s}_\theta$ to approximate the term $\nabla_{\mathbf{x}_t} \log p_t(\mathbf{x}_t)$. The term $p_t(\mathbf{x}_{\text{ADV}} \mid \mathbf{x}_t)$ is modeled according to equation 10, as before assuming it follows a Gaussian distribution with a mean of the one-step Euler prediction $\hat{\mathbf{x}}_0$. Note that ScoreAG, just as other purification methods, cannot detect adversarial images. Therefore, it also needs to preserve image semantics if there is no perturbation.

## 4 Experimental Evaluation

The primary objective of our experimental evaluation is to assess the capability of ScoreAG in generating and purifying adversarial examples. More specifically, we investigate the following properties of ScoreAG: **(1)** the ability to synthesize adversarial examples from scratch (GAS), **(2)** the ability to transform existing images into adversarial examples (GAT), and **(3)** the enhancement of classifier robustness by leveraging the generative capability of the model to purify images (GAP). This evaluation aims to provide comprehensive insights into the strengths and limitations of ScoreAG in the realm of adversarial example generation and classifier robustness.

**Baselines.** In our evaluation, we benchmark our adversarial attacks against a wide range of established methods covering various threat models. Specifically, we consider the fast gradient sign-based approaches FGSM (Goodfellow et al., 2014), DI-FGSM (Xie et al., 2019), and SI-NI-FGSM (Lin et al., 2019). In addition, we include Projected Gradient Descent-based techniques, specifically Adaptive Projected Gradient Descent (APGD) and its targeted variant (APGDT) (Croce & Hein, 2020b). For a comprehensive assessment, we also examine single pixel, black-box, and minimal perturbation methods, represented by OnePixel (Su et al., 2019), Square (Andriushchenko et al., 2020) and Fast Adaptive Boundary (FAB) (Croce & Hein, 2020a), respectively. Finally, we compare to the unrestricted attacks Composite Adversarial Attack (CAA) (Hsiung et al., 2023), PerceptualPGDAttack (PPGD), FastLagrangePerceptualAttack (LPA) (Laidlaw et al., 2020), and DiffAttack (Chen et al., 2023a), which is based on latent diffusion. Furthermore, we compare to Adversarial Content Attack (ACA) (Chen et al., 2023c) in App. B.3.[1]

To evaluate the efficacy of ScoreAG in purifying adversarial examples, we conduct several experiments in a preprocessor-blackbox setting. For the evaluation, we employ the targeted APGDT and untargeted APGD attacks (Croce & Hein, 2020b) and ScoreAG in the GAS setup. Our experiments also incorporate the purifying methods ADP (Yoon et al., 2021) and DiffPure (Nie et al., 2022). Additionally, we compare with state-of-the-art adversarial training techniques that partially utilize supplementary data from generative models (Cui et al., 2023; Wang et al., 2023; Peng et al., 2023).

**Experimental Setup.** We employ three benchmark datasets for our experiments: CIFAR10, CIFAR100 (Krizhevsky et al., 2009), and TinyImagenet. We utilize pre-trained Elucidating Diffusion Models (EDM) in the variance preserving (VP) setup (Karras et al., 2022; Wang et al., 2023) for image generation. As our classifier, we opt for the well-established WideResNet architecture WRN-28-10 (Zagoruyko & Komodakis, 2016). The classifiers are trained for 400 epochs using SGD with Nesterov momentum of 0.9 and weight decay of $5 \times 10^{-4}$. Additionally, we incorporate a cyclic learning rate scheduler with cosine annealing (Smith

---

[1]We do not compare to Chen et al. (2023b) as their code was not publicly available upon submission and multiple attempts to contact the authors were unsuccessful.

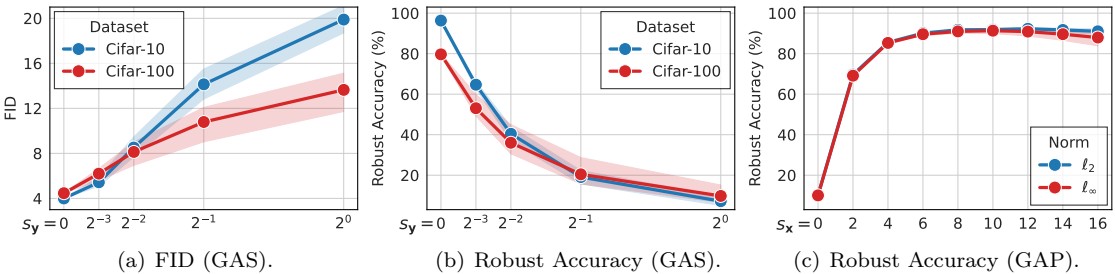

(a) FID (GAS).  (b) Robust Accuracy (GAS).  (c) Robust Accuracy (GAP).

Figure 3: FID (a) and accuracy (b) for increasing $s_\mathbf{y}$ scales in the synthesis (GAS) setup, and robust accuracy (c) for increasing $s_\mathbf{x}$ scales in the purification (GAP) setup under APGD attack. Classifier: WRN-28-10. The shaded area shows the 95% CI over four seeds.

& Topin, 2019) with an initial learning rate of 0.2. To further stabilize the training process, we apply exponential moving average with a decay rate of 0.995. Each classifier is trained four times to ensure the reproducibility of our results, and we report standard deviations with ($\pm$). For pretrained classifiers with only one available model, we do not report standard deviations. For the restricted methods, we consider the common norms in the literature $\ell_2 = 0.5$ for CIFAR10 and CIFAR100, $\ell_2 = 2.5$ for TinyImagenet, and $\ell_\infty = 8/255$ for all three datasets. For DiffAttack, ACA, and DiffPure we take the implementation of the official repositories, while we use Torchattacks (Kim, 2020) for the remaining baselines. The runtimes for all methods are shown in Tab. 11 in the appendix.

**Evaluation Metrics.** To evaluate our results, we compute the robust accuracy, i.e., the accuracy after an attack. Furthermore, we use the clean accuracy, i.e., the accuracy of a (robust) model without any attack. For the GAS task, we use the FID (Heusel et al., 2017) to assess the similarity between the distribution of synthetic images and the test set, providing a distribution-level measure. Since FID is not suitable for instance-based evaluation, we use the LPIPS score (Zhang et al., 2018) for the GAT task to measure perceptual similarity at the instance level.

## 4.1 Quantitative Results

**Evaluating Generative Adversarial Synthesis.** As explained in Sec. 3.2, ScoreAG is capable of synthesizing adversarial examples. Fig. 3(a) and Fig. 3(b) show the accuracy and the FID of a WRN-28-10 classifier as $s_\mathbf{y}$ increases, respectively. Notably, the classifier yields nearly identical performance as on real data when $s_\mathbf{y} = 0$. However, even a minor increase of $s_\mathbf{y}$ to 0.125 results in a substantial reduction in accuracy while maintaining a low FID. Setting $s_\mathbf{y}$ to 1.0 causes the classifier's performance to drop below random guessing levels for the CIFAR10 dataset. Additionally, Fig. 4(a) presents sample images generated at various scales. Notably, increasing $s_\mathbf{y}$ leads to subtle modifications in the images. Rather than introducing random noise, these changes maintain image coherence up to a scale of $s_\mathbf{y} = 0.5$. Beyond this point, specifically at $s_\mathbf{y} = 1.0$, there is a noticeable decline in image quality, as reflected by the FID.

Since our approach leverages a generative model, it enables the synthesis of an unlimited number of adversarial examples, thereby providing a more comprehensive robustness assessment. Moreover, in scenarios requiring the generation of adversarial examples, our method allows for rejection sampling at low $s_\mathbf{y}$ scales, ensuring the preservation of image quality. This is particularly important for adversarial training, where synthetic images can enhance robustness (Wang et al., 2023).

**Evaluating Generative Adversarial Transformation.** Beyond the synthesis of new adversarial examples, our framework allows converting pre-existing images into adversarial ones as described in Sec. 3.3. We show the accuracies and LPIPS scores of various attacks in Tab. 1. Notably, ScoreAG consistently achieves 0% accuracy, lower than the $\ell_2$ and $\ell_0$ restricted methods across all three datasets, making it competitive to APGDT and LPA. This demonstrates ScoreAG's capability of generating adversarial examples. Surprisingly, the other unrestricted diffusion-based method, DiffAttack, yields considerably lower attack success rates. We attribute this discrepancy to the fact that it only leverages the last few iterations of the denoising diffusion process. Finally, we observe that the LPIPS scores of ScoreAG are comparable to the restricted meth-

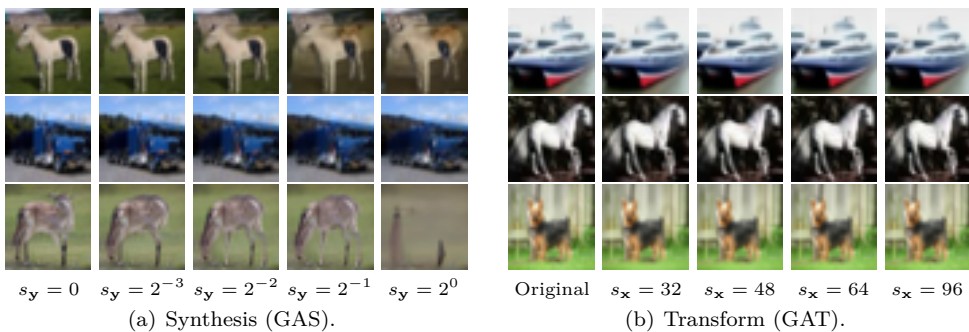

| $s_{\mathbf{y}} = 0$ | $s_{\mathbf{y}} = 2^{-3}$ | $s_{\mathbf{y}} = 2^{-2}$ | $s_{\mathbf{y}} = 2^{-1}$ | $s_{\mathbf{y}} = 2^0$ | Original | $s_{\mathbf{x}} = 32$ | $s_{\mathbf{x}} = 48$ | $s_{\mathbf{x}} = 64$ | $s_{\mathbf{x}} = 96$ |

(a) Synthesis (GAS).      (b) Transform (GAT).

Figure 4: Examples on the CIFAR10 dataset. Fig. 4(a) shows the synthesis (GAS) setup and generates images of the classes "horse", "truck", and "deer", which are classified as "automobile", "ship", and "horse", respectively, as $s_{\mathbf{y}}$ increases. Fig. 4(b) shows the transformation (GAT) setup and transforms images of the classes "ship", "horse", and "dog", into adversarial examples classified as "ship", "deer", and "cat". For $s_{\mathbf{x}} = 32$, the images are outside of common perturbation norms, i.e., $\ell_2 = 0.5$ and $\ell_\infty = 8/255$, but preserve image semantics. We show examples of selected baselines in Fig. 5.

Table 1: Robust accuracy and LPIPS scores for various attacks on CIFAR10, CIFAR100, and TinyImagenet. Best scores are in **bold**, second best underlined.

| Dataset | Robust Accuracy in % (↓) | | | LPIPS (↓) | | |
|---|---|---|---|---|---|---|
| | CIFAR10 | CIFAR100 | TinyImagenet | CIFAR10 | CIFAR100 | TinyImagenet |
| **$\ell_\infty$ restricted** | | | | | | |
| FGSM (Goodfellow et al., 2014) | $31.47_{\pm 13.39}$ | $10.82_{\pm 1.62}$ | $1.42_{\pm 0.17}$ | $30.27_{\pm 1.41}$ | $39.44_{\pm 1.45}$ | $180.76_{\pm 2.27}$ |
| DI-FGSM (Xie et al., 2019) | $0.54_{\pm 0.54}$ | $0.13_{\pm 0.10}$ | $0.04_{\pm 0.02}$ | $18.98_{\pm 3.63}$ | $22.87_{\pm 2.48}$ | $125.46_{\pm 3.75}$ |
| SI-NI-FGSM (Lin et al., 2019) | $3.01_{\pm 0.93}$ | $1.20_{\pm 0.16}$ | $0.69_{\pm 0.11}$ | $29.57_{\pm 3.69}$ | $40.92_{\pm 4.84}$ | $156.27_{\pm 3.49}$ |
| APGD (Croce & Hein, 2020b) | $0.18_{\pm 0.21}$ | $\underline{0.10_{\pm 0.03}}$ | $0.18_{\pm 0.03}$ | $12.40_{\pm 1.64}$ | $12.52_{\pm 1.52}$ | $88.10_{\pm 2.20}$ |
| APGDT (Croce & Hein, 2020b) | $\mathbf{0.00_{\pm 0.00}}$ | $\mathbf{0.00_{\pm 0.00}}$ | $\mathbf{0.00_{\pm 0.00}}$ | $12.17_{\pm 0.11}$ | $11.56_{\pm 0.53}$ | $65.64_{\pm 1.55}$ |
| Square (Andriushchenko et al., 2020) | $0.25_{\pm 0.24}$ | $0.19_{\pm 0.04}$ | $0.51_{\pm 0.05}$ | $126.20_{\pm 1.61}$ | $88.43_{\pm 0.9}$ | $127.93_{\pm 1.46}$ |
| FAB (Croce & Hein, 2020a) | $1.67_{\pm 1.56}$ | $0.76_{\pm 0.06}$ | $0.11_{\pm 0.19}$ | $0.78_{\pm 0.36}$ | $\underline{0.15_{\pm 0.01}}$ | $7.05_{\pm 0.09}$ |
| **$\ell_2$ restricted** | | | | | | |
| APGD (Croce & Hein, 2020b) | $1.21_{\pm 0.05}$ | $0.69_{\pm 0.01}$ | $0.15_{\pm 0.05}$ | $2.48_{\pm 0.18}$ | $2.66_{\pm 0.18}$ | $96.01_{\pm 2.92}$ |
| APGDT (Croce & Hein, 2020b) | $0.11_{\pm 0.01}$ | $0.09_{\pm 0.01}$ | $\mathbf{0.00_{\pm 0.00}}$ | $2.51_{\pm 0.1}$ | $2.33_{\pm 0.17}$ | $101.13_{\pm 2.62}$ |
| Square (Andriushchenko et al., 2020) | $19.67_{\pm 0.27}$ | $7.02_{\pm 0.42}$ | $1.26_{\pm 0.10}$ | $8.54_{\pm 0.13}$ | $10.06_{\pm 0.58}$ | $151.46_{\pm 2.58}$ |
| FAB (Croce & Hein, 2020a) | $7.41_{\pm 6.19}$ | $1.44_{\pm 0.33}$ | $\underline{0.01_{\pm 0.01}}$ | $\mathbf{0.36_{\pm 0.06}}$ | $\mathbf{0.10_{\pm 0.01}}$ | $\mathbf{0.56_{\pm 0.06}}$ |
| **$\ell_0$ restricted** | | | | | | |
| OnePixel (Su et al., 2019) | $82.82_{\pm 0.94}$ | $59.17_{\pm 0.77}$ | $59.42_{\pm 0.38}$ | $10.67_{\pm 1.16}$ | $13.65_{\pm 0.54}$ | $11.08_{\pm 0.10}$ |
| **Unrestricted** | | | | | | |
| CAA (Hsiung et al., 2023) | $43.23_{\pm 0.71}$ | $12.88_{\pm 0.41}$ | $8.81_{\pm 0.44}$ | $1564.62_{\pm 23.23}$ | $1266.86_{\pm 22.07}$ | $879.52_{\pm 14.94}$ |
| PPGD (Laidlaw et al., 2020) | $31.82_{\pm 2.77}$ | $39.76_{\pm 2.08}$ | $2.76_{\pm 0.10}$ | $10.70_{\pm 0.12}$ | $7.23_{\pm 0.18}$ | $31.13_{\pm 0.63}$ |
| LPA (Laidlaw et al., 2020) | $\underline{0.04_{\pm 0.05}}$ | $\mathbf{0.00_{\pm 0.00}}$ | $\mathbf{0.00_{\pm 0.00}}$ | $25.41_{\pm 6.50}$ | $40.08_{\pm 9.57}$ | $339.48_{\pm 6.03}$ |
| DiffAttack (Chen et al., 2023a) | $14.40_{\pm 0.97}$ | $4.89_{\pm 1.57}$ | $2.13_{\pm 0.09}$ | $637.89_{\pm 3.68}$ | $626.99_{\pm 4.98}$ | $808.90_{\pm 6.36}$ |
| ScoreAG (Ours) | $\mathbf{0.00_{\pm 0.00}}$ | $\mathbf{0.00_{\pm 0.00}}$ | $\mathbf{0.00_{\pm 0.00}}$ | $4.39_{\pm 0.13}$ | $4.28_{\pm 0.22}$ | $109.11_{\pm 0.55}$ |
| ScoreAG-LPIPS (Ours) | $\mathbf{0.00_{\pm 0.00}}$ | $0.01_{\pm 0.01}$ | $\mathbf{0.00_{\pm 0.00}}$ | $\underline{0.63_{\pm 0.03}}$ | $0.54_{\pm 0.02}$ | $42.83_{\pm 3.39}$ |

ods, and competitive to the minimum perturbation method FAB when applying additional LPIPS guidance (ScoreAG-LPIPS), demonstrating its semantic preserving property. We present results on more classifiers in Tab. 8 and on a high-resolution dataset in Tab. 9 in the appendix. In Tab. 2, we show the accuracies of APGD, APGDT, and ScoreAG on robust classifiers. Notably, ScoreAG demonstrates a considerably superior attack success rate compared to the PGD-based attacks. We attribute this to the more comprehensive robustness assessment of ScoreAG. While most baselines only assess the robustness of adversarial examples on the $\ell_p$-constraint border, ScoreAG draws samples from the distribution of semantics-preserving adversarial examples (see Sec. 3.3).

**Evaluating Generative Adversarial Purification.** Finally, we examine the purification ability of ScoreAG. Tab. 2 shows the purification results for various methods on the CIFAR10 dataset. Our results show that ScoreAG consistently achieves state-of-the-art performance in robust accuracy, outperforming other adversarial purification and training methods. Notably, ScoreAG not only successfully defends attacks but also maintains a high level of clean accuracy comparable to that of adversarial training. This demonstrates ScoreAG's capability to preserve the core semantics while effectively neutralizing the impact of adversarial perturbations by projecting samples on the manifold of clean images. Overall, our results indicate that purification methods can defend better against adversarial attacks than adversarial training approaches, which we attribute to the preprocessor-blackbox setting. Note that it is not possible to detect

Table 2: CIFAR10 robust accuracy of different adversarial training and purification methods for the attacks APGD, APGDT, and ScoreAG. If multiple threat models exist, we denote results as $\ell_\infty/\ell_2$. Best purification scores are in **bold**, best attack success rates are underlined.

| Model | Clean Accuracy | APGD | | APGDT | | ScoreAG-GAT (Ours) Unrestricted | Architecture |
|---|---|---|---|---|---|---|---|
| | | $\ell_\infty$ | $\ell_2$ | $\ell_\infty$ | $\ell_2$ | | |
| **Adversarial Training** | | | | | | | |
| (Cui et al., 2023) | 92.16 | 70.36 | - | 68.43 | - | 47.69 | WRN-28-10 |
| (Wang et al., 2023) | 92.44 / 95.16 | 70.08 | 84.52 | 68.04 | 83.88 | 45.33 / 38.49 | WRN-28-10 |
| (Wang et al., 2023) | 93.25 / 95.54 | 73.29 | 85.65 | 71.42 | 85.28 | 41.52 / 41.37 | WRN-70-16 |
| (Peng et al., 2023) | 93.27 | 73.67 | - | 71.82 | - | 38.87 | RaWRN-70-16 |
| **Adversarial Purification** | | | | | | | |
| ADP (Yoon et al., 2021) | 93.09 | - | - | 85.45 | - | - | WRN-28-10 |
| DiffPure (Nie et al., 2022) | 89.02 | 87.72 | 88.46 | 88.30 | 88.18 | 88.57 | WRN-28-10 |
| ScoreAG-GAP (Ours) | 93.93±0.12 | **91.34±0.46** | **92.13±1.41** | **90.25±0.44** | **90.89±0.40** | **90.74±0.67** | WRN-28-10 |

adversarial examples. Therefore, the purification needs to be applied to all images. However, ScoreAG still achieves a high clean accuracy. We demonstrate its applicability to common corruptions in App. B.4.

**Hyperparameter study.** We explore the impact of the scale parameters $s_\mathbf{y}$ and $s_\mathbf{x}$ on accuracy and FID, as depicted in Fig. 3. In Fig. 3(c), we examine the efficacy of purification against adversarial attacks of APGD under both $\ell_2$ and $\ell_\infty$ norms across different $s_\mathbf{x}$ scales. At $s_\mathbf{x} = 0$, the generated images are unconditional without guidance and independent of the input. Therefore, the robust accuracy equals random guessing. As $s_\mathbf{x}$ increases, the accuracy improves, reaching a performance plateau at approximately $s_\mathbf{x} = 10$. Increasing $s_\mathbf{x}$ further reduces the accuracy as the sampled images start to resemble adversarial perturbations. In practice, we scale $s$ by $t^{-1}$.

Finally, Tab. 3 shows the robust accuracy and median $\ell_2$ distances across different scale configurations for the CIFAR10 and CIFAR100 datasets. We can observe that an increase in $s_\mathbf{y}$ leads to reduced classifier accuracy for CIFAR10, improving the efficacy of the adversarial attacks. A rise in $s_\mathbf{x}$, however, increases the accuracy as the generated image closer resembles the original. The median $\ell_2$ distance exhibits a similar behavior. While a lower $s_\mathbf{y}$ yields no difference for both datasets, increasing $s_\mathbf{x}$ decreases the median distances for CIFAR10 and CIFAR100. In Fig. 4(b), we show examples across various $s_\mathbf{x}$ scales on the CIFAR10 dataset. Notably, all scales preserve the image semantics and do not display any observable differences. In practice, we iteratively increase the scale $s_\mathbf{y}$ if the attack is not successful.

Table 3: Robust accuracy and median $\ell_2$ distances for various hyperparameter configurations. Best scores are in **bold**.

| Dataset | Robust Accuracy in % (↓) | | Median $\ell_2$ distance | |
|---|---|---|---|---|
| | CIFAR10 | CIFAR100 | CIFAR10 | CIFAR100 |
| $\mathbf{s_y = 48}$ | | | | |
| $s_\mathbf{x} = 16$ | **0.10** | **0.02** | 1.12 | 1.09 |
| $s_\mathbf{x} = 32$ | 0.23 | **0.02** | 0.65 | 0.64 |
| $s_\mathbf{x} = 48$ | 0.32 | 0.03 | 0.49 | 0.49 |
| $s_\mathbf{x} = 64$ | 0.34 | 0.04 | 0.43 | 0.40 |
| $\mathbf{s_y = 64}$ | | | | |
| $s_\mathbf{x} = 48$ | 0.22 | 0.17 | 0.50 | 0.49 |
| $s_\mathbf{x} = 64$ | 0.24 | **0.02** | 0.43 | 0.40 |
| $s_\mathbf{x} = 96$ | 0.28 | 0.03 | 0.35 | 0.30 |
| $\mathbf{s_y = 96}$ | | | | |
| $s_\mathbf{x} = 48$ | **0.10** | 0.17 | 0.51 | 0.50 |
| $s_\mathbf{x} = 64$ | 0.11 | 0.21 | 0.44 | 0.40 |
| $s_\mathbf{x} = 96$ | 0.13 | 0.34 | 0.35 | 0.30 |

## 4.2 Qualitative Analysis

To investigate the quality of the adversarial attacks, we deploy ScoreAG on the ImageNet dataset (Deng et al., 2009) with a resolution of $256 \times 256$. We use the latent diffusion model DiT proposed by Peebles & Xie (2022), along with a pre-trained latent classifier from (Kim et al., 2022). The images are sampled using the denoising procedure by Kollovieh et al. (2023) as explained in Sec. 3.2. Note that as the generative process is performed in the latent space, the model has more freedom in terms of reconstruction.

We show an example image of a tiger shark in Fig. 1 with corresponding adversarial attacks. While the classifier correctly identifies the tiger shark in the baseline image, it fails to do so in the generated adversarial examples. Notably, the $\ell_p$-bounded methods display noticeable noisy fragments. In contrast, ScoreAG produces clean adversarial examples, altering only minor details while retaining the core semantics — most notably, the removal of a small fish — which prove to be important classification cues. We provide further examples for GAS in Sec. B.6 and for GAT in Sec. B.7. The synthetic images display a high degree of realism, and the transformed images show visible differences while preserving the semantics of the original image.

### 4.3 Human Study

To evaluate whether ScoreAG generates semantics-preserving adversarial examples, we perform a human study on adversarially modified (real) as well as synthetically generated images. For the study, we choose CIFAR10 images as it (1) avoids any class-selection bias, whereas high-resolution datasets usually contain many classes only distinguishable by human experts; and (2) is the most commonly used dataset in related work. Hyperparameters are set to produce an interesting regime, where the generated adversarial images are significantly outside common $\ell_p$-norm balls and constitute strong attacks for the classifier in question. In particular, we randomly sample five images from each class to generate 50 adversarial examples using $s_{\mathbf{x}} = 16$ and $s_{\mathbf{y}} = 48$. These adversarial examples have an average $\ell_2$-norm difference to their clean counterparts of $0.68 \pm 0.24$, exceeding the common $\ell_2$-norm ball constraint of 0.5 (Croce et al., 2020) by on average 36%. For the synthetic examples, we generate 50 images without ($s_{\mathbf{y}} = 0$) and 50 images with guidance ($s_{\mathbf{y}} = 0.125$), again in a class-balanced fashion. For the adversarial guided synthetic examples, we employ rejection sampling to only consider images that lead to misclassification by the classifier. To ensure high data quality for the study, we used the Prolific platform (Eyal et al., 2021) to employ 60 randomly chosen human evaluators to label the 200 images. To avoid bias, we presented the adversarial examples (synthetic or modified) before the unperturbed examples and introduced the category "Other / I don't know".

We compute human accuracy by choosing the majority vote class of all 60 human evaluators and compare it with the ground truth class. We show the results of the human study in Tab. 4. Notably, humans can still accurately classify 94% of the adversarial modified images despite significantly larger $\ell_2$ distances, establishing *almost perfect* semantic preservation for GAT. For GAS, humans classify 70% of the (successful) synthetic adversarial images correctly. This is lower than for adversarial modification and shows that the generation of completely synthetic semantics-preserving adversarial

Table 4: Human study to evaluate the adversarial examples of ScoreAG. The human ACC corresponds to the majority vote.

| Dataset | Model ACC | Human ACC |
|---|---|---|
| **Clean** | | |
| Real | 98% | 100% |
| Synthetic | 94% | 94% |
| **Adversarial** | | |
| Real | 2% | 94% |
| Synthetic | 0% | 70% |

examples is a harder task than adversarial modification. Still, GAS achieves *good* semantic preservation, significantly outperforming random guessing (10%). We believe it is critical that semantic preservation of unrestricted attacks is evaluated through human studies as done in some early works (Song et al., 2018; Khoshpasand & Ghorbani, 2020). As this is missing in all related unrestricted attack works used as baselines in this work, we hope to contribute to establishing this as an evaluation standard, and that our results can serve as interesting baselines for future works.

## 5 Related Work

**Diffusion Models.** Diffusion models (Sohl-Dickstein et al., 2015; Ho et al., 2020) and score-based generative models (Song et al., 2020) received significant attention in recent years, owing to their remarkable performance across various domains (Kong et al., 2020; Lienen et al., 2023; Kollovieh et al., 2023) and have since emerged as the go-to methodology for many generative tasks. Dhariwal & Nichol (2021) proposed diffusion guidance to perform conditional sampling using unconditional models. A recent study has shown that classifiers can enhance their robust accuracy when training on images generated by diffusion models (Wang et al., 2023), demonstrating the usefulness and potential of diffusion models in the robustness domain.

**Adversarial Attacks.** An important line of work are white-box approaches, which have full access to the model parameters and gradients, such as the fast gradient sign method (FGSM) introduced by Goodfellow et al. (2014). While FGSM and its subsequent extensions (Xie et al., 2019; Dong et al., 2018; Lin et al., 2019; Wang, 2021) primarily focus on perturbations constrained by the $\ell_\infty$ norm, other white-box techniques employ projected gradient descent and explore a broader range of perturbation norms (Madry et al., 2017; Zhang et al., 2019). In contrast, black-box attacks are closer to real-world scenarios and do not have access to model parameters or gradients (Narodytska & Kasiviswanathan, 2016; Brendel et al., 2017; Andriushchenko et al., 2020). As ScoreAG-GAT and ScoreAG-GAS rely on the gradients of the classifier to compute guidance scores, they are categorized as white-box attacks.

**Diffusion-Based Attacks.** Two recent works by Chen et al. (2023a) and Xue et al. (2023) propose DiffAttack and Diff-PGD, respectively. Diff-PGD performs projected gradient descent in the latent diffusion space to obtain $\ell_\infty$-bounded adversarial examples, whereas DiffAttack generates *unrestricted* adversarial examples by leveraging a latent diffusion model. However, as both methods employ only the final denoising stages of the diffusion process in a similar fashion to SDEdit (Meng et al., 2021), the adversarial perturbations only incorporate changes of high-level features. Finally, Chen et al. (2023c) implement PGD in the $\ell_\infty$-norm within the latent space of stable diffusion. In parallel, Chen et al. (2023b) apply PGD iteratively at each step of the diffusion process and combine it with adversarial inpainting. Unlike previous works, ScoreAG does not rely on PGD in the latent space for its attack and semantic preservation, but solely leverages the diffusion manifold in combination with a task-specific guidance.

**Adversarial Purification.** In response to the introduction of adversarial attacks, a variety of adversarial purification methods to defend machine learning models have emerged. Early works utilized Generative Adversarial Networks (GANs) Song et al. (2017; 2018); Samangouei et al. (2018) and Energy-Based Models (EBMs) (Hill et al., 2020) to remove adversarial perturbations from images. More recent methods have shifted focus towards score-based generative models, like ADP (Yoon et al., 2021), and diffusion models, such as DiffPure (Nie et al., 2022). However, ADP and DiffPure only denoise with small noise magnitudes during the purification process and are thereby limited to correcting high-level adversarial features, whereas ScoreAG traverses the whole diffusion process, providing more flexibility in purifying perturbations. Kang et al. (2023) have recently shown that these purification methods decrease in effectiveness in a white-box setting by evasion attacks. However, as previously mentioned, we focus on preprocessor black-box attacks, which are more relevant in real-world problems.

# 6 Discussion

**Limitations and Future Work.** Our work demonstrates the potential and capabilities of score-based generative models in the realm of adversarial attacks and robustness. While ScoreAG is able to generate and purify adversarial attacks, some drawbacks remain. Primarily, the evaluation of unrestricted attacks remains challenging. We resolve this limitation by performing a human study and argue that this should become standard. Moreover, the proposed purification approach is only applicable to a preprocessor-blackbox setting, as computing the gradients of the generative process efficiently is an open problem.

**Conclusion.** In this work, we address the question of how to generate *unrestricted* adversarial examples. We introduce ScoreAG, a novel framework that bridges the gap between adversarial attacks and score-based generative models. Utilizing diffusion guidance and pre-trained models, ScoreAG can synthesize new adversarial attacks, transform existing images into adversarial examples, and purify images, thereby enhancing the empirical robust accuracy of classifiers. Our results indicate that ScoreAG can effectively generate semantics-preserving adversarial images beyond the limitations of the $\ell_p$-norms. Our experimental evaluation demonstrates that ScoreAG matches the performance of existing state-of-the-art attacks and defenses. We see unrestricted adversarial examples - as generated by our work - as vital to achieve a holistic view of robustness and complementary to hand-picked common corruptions (Kar et al., 2022) or classical $\ell_p$ threat models.

**Broader Impact** This work contributes to the domain of robustness, focusing on unrestricted adversarial attacks. Our framework, ScoreAG, is designed for the generation and purification of adversarial images. While there exists the potential for malicious misuse, we hope for our insights to enhance the understanding of machine learning models' robustness. Moreover, despite the competitive empirical performance of ScoreAG, we advise against relying solely on the algorithm.

# Acknowledgments

This paper has been supported by the Munich Center for Machine Learning and by the DAAD program Konrad Zuse Schools of Excellence in Artificial Intelligence, sponsored by the German Federal Ministry of Education and Research, and the German Research Foundation, grant GU 1409/4-1.

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

# A  Experimental Setup and Hyperparameters

## A.1  Reproducibility

Our models are implemented using PyTorch with the pre-trained EDM models by Karras et al. (2022) and Wang et al. (2023), and the guidance scores are computed using automatic differentiation. In Tab. 5 and Tab. 6, we give an overview of the hyperparameters of ScoreAG. For the methods DiffAttack, DiffPure, CAA, PPGD, and LPA, we use the corresponding authors' official implementations with the suggested hyperparameters. For the remaining attacks, we use Adversarial-Attacks-PyTorch with its default parameters (Kim, 2020).

## A.2  Hyperparameters

To train the WRN-28-10 classifiers, we use the parameters shown in Tab. 5. In Tab. 6, we show the scale parameters used to evaluate the attacks and purification of ScoreAG, i.e., the results shown in Tab. 1, Tab. 2 and Tab. 8. The attacks on robust models do not sequentially increase the scale $s_{\mathbf{y}}$ but use fixed scales of $s_{\mathbf{x}} = 48$ and $s_{\mathbf{y}} = 80$. For the common corruptions we use a scale of $s_{\mathbf{x}} = 40$ on the robust models. Finally, for the EDM sampler we use the default sampling scheduler and parameters by Karras et al. (2022).

| Hyperparameter | Value |
|---|---|
| Number of epochs | 400 |
| Optimizer | SGD |
| Nesterov momentum | 0.9 |
| Weight decay | $5 \times 10^{-4}$ |
| Exponential moving average | 0.995 |
| Learning rate scheduler | Cyclic with cosine annealing |
| Initial learning rate | 0.2 |

Table 5: Hyperparameters used to train the WRN-28-10 classifiers.

| Hyperparameter | Value |
|---|---|
| **CIFAR10** | |
| $s_{\mathbf{y}}$ (GAT) | 32 |
| $s_{\mathbf{x}}$ (GAT) | 48 |
| $s_{\mathbf{y}}$ (GAT-LPIPS) | 32 |
| $s_{\mathbf{x}}$ (GAT-LPIPS) | 48 |
| $s_{\mathrm{LPIPS}}$ (GAT-LPIPS) | 48 |
| $s_{\mathbf{x}}$ (GAP) | 10 |
| increments (GAT) | 20 |
| steps (GAP) | 72 |
| steps (GAT) | 512 |
| **CIFAR100** | |
| $s_{\mathbf{y}}$ (GAT) | 32 |
| $s_{\mathbf{x}}$ (GAT) | 48 |
| $s_{\mathbf{y}}$ (GAT-LPIPS) | 32 |
| $s_{\mathbf{x}}$ (GAT-LPIPS) | 48 |
| $s_{\mathrm{LPIPS}}$ (GAT-LPIPS) | 48 |
| increments | 20 |
| steps | 512 |
| **TinyImagenet** | |
| $s_{\mathbf{y}}$ (GAT) | 64 |
| $s_{\mathbf{x}}$ (GAT) | 16 |
| increments | 20 |
| steps | 512 |
| **Imagenet-Compatible** | |
| $s_{\mathbf{y}}$ (GAT) | 8 |
| $s_{\mathbf{x}}$ (GAT) | 0.5 |
| increments | 4 |
| steps | 1000 |

Table 6: Hyperparameters used to evaluate ScoreAG.

| Hyperparameter | Value |
|---|---|
| $\sigma_{\min}$ | 0.002 |
| $\sigma_{\max}$ | 80 |
| $\rho$ | 7 |
| $S_{\text{churn}}$ | 0/4 |
| $S_{\text{noise}}$ | 1 |

Table 7: Hyperparameters used for sampling using Alg. 15.

## A.3 Pseudocode

We present the pseudocode of ScoreAG in Alg. 15, implementing the sampler proposed by Karras et al. (2022). Here, $s$ denotes the scale parameter for the task, while $t_i$ and $\gamma_i$ are scheduler parameters retained from the original configuration (see Tab. 7). More specifically, $\gamma_i = \min(S_{\text{churn}}, \sqrt{2} - 1)$ and

$$t_{i<N} = \left( \sigma_{\max}^{\frac{1}{\rho}} + \frac{i}{N-1} \left( \sigma_{\min}^{\frac{1}{\rho}} - \sigma_{\max}^{\frac{1}{\rho}} \right) \right)^{\rho}, \quad t_N = 0. \tag{12}$$

We compute the different guidance scores using equation 6, equation 9, and equation 11.

---

**Algorithm 1** ScoreAG with the sampler of Karras et al. (2022).

1: **procedure** SCOREAG($s_\theta(\mathbf{x}; \sigma), t_{i \in \{0,\dots,N\}}, \gamma_{i \in \{0,\dots,N-1\}}, s, c$)
2:      sample $\mathbf{x}_0 \sim \mathcal{N}(0, t_0^2 \mathbf{I})$
3:      **for** $i \in \{0, \dots, N-1\}$ **do**
4:          sample $\epsilon_i \sim \mathcal{N}(0, S_{\text{noise}}^2 \mathbf{I})$
5:          $\hat{t}_i \leftarrow t_i + \gamma_i t_i$
6:          $\hat{\mathbf{x}}_i \leftarrow \mathbf{x}_i + \sqrt{\hat{t}_i^2 - t_i^2} \, \epsilon_i$
7:          $d_i \leftarrow \hat{t}_i \cdot \left( s_\theta(\mathbf{x}_i, \hat{t}_i) + s \cdot \nabla_{\mathbf{x}_i} \log p_{\hat{t}_i}(c \mid \mathbf{x}_i) \right)$
8:          $\mathbf{x}_{i+1} \leftarrow \hat{\mathbf{x}}_i + (t_{i+1} - \hat{t}_i) d_i$
9:          **if** $t_{i+1} \neq 0$ **then**
10:             $d_i' \leftarrow t_{i+1} \cdot \left( s_\theta(\mathbf{x}_{i+1}, t_{i+1}) + s \cdot \nabla_{\mathbf{x}_{i+1}} \log p_{t_{i+1}}(c \mid \mathbf{x}_{i+1}) \right)$
11:             $\mathbf{x}_{i+1} \leftarrow \hat{\mathbf{x}}_i + (t_{i+1} - \hat{t}_i) \left( \frac{1}{2} d_i + \frac{1}{2} d_i' \right)$
12:          **end if**
13:      **end for**
14:      **return** $\mathbf{x}_N$
15: **end procedure**

---

# B    Additional Results

## B.1    Qualitative Comparison of Baselines

In Fig. 5, we visualize adversarial attacks of selected baselines for the images in Fig. 4.

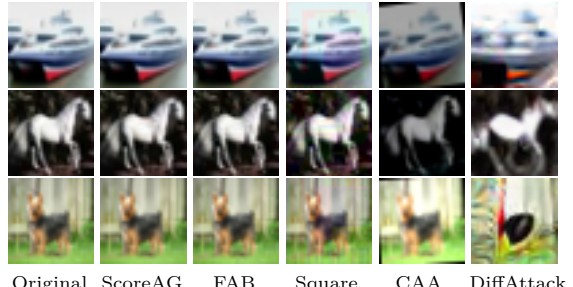

Original   ScoreAG   FAB   Square   CAA   DiffAttack

Figure 5: Examples from the CIFAR10 dataset. The figure presents selected baseline images corresponding to the examples in Fig. 4(b). For ScoreAG-GAT, we used $s_{\mathbf{x}} = 48$. As baselines, we included FAB ($\ell_2 = 0.5$) and Square ($\ell_\infty = 8/255$) to represent restricted attacks, as they achieve the lowest and highest LPIPS scores, respectively. Additionally, we show the two unrestricted baselines, CAA and DiffAttack.

## B.2    Qualitative Effect of the Scale Parameters

To provide a more intuitive understanding of ScoreAG, we show the visual effect of the scale parameters $s_{\mathbf{x}}$ and $s_{\mathbf{y}}$ in Fig. 6 and 7. These visualizations illustrate the effects of the scale parameters $s_{\mathbf{x}}$ and $s_{\mathbf{y}}$. When both scale parameters are set to zero, the model behaves as a standard diffusion model. Increasing $s_{\mathbf{x}}$ guides the diffusion process toward a specific image, which is used in the GAP setup. Increasing $s_{\mathbf{y}}$ introduces adversarial perturbations, allowing the synthesis of adversarial images. When both parameters are greater than zero, the GAT model transforms existing images into adversarial examples.

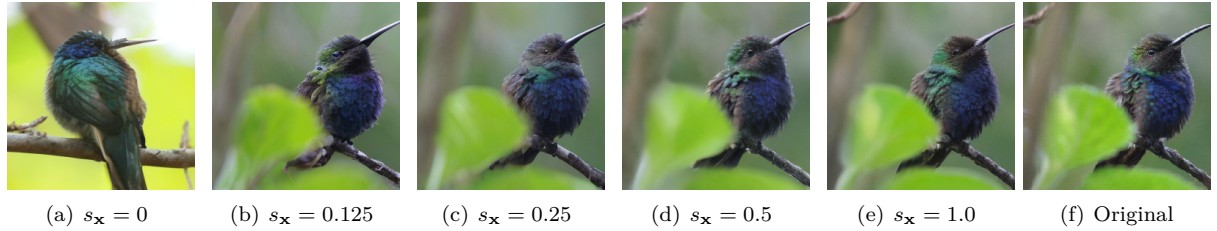

(a) $s_{\mathbf{x}} = 0$      (b) $s_{\mathbf{x}} = 0.125$      (c) $s_{\mathbf{x}} = 0.25$      (d) $s_{\mathbf{x}} = 0.5$      (e) $s_{\mathbf{x}} = 1.0$      (f) Original

Figure 6: Effect of the scale parameter $s_{\mathbf{x}}$. The images display adversarial images generated by ScoreAG-GAT across different scales $s_{\mathbf{x}}$ on a robust WRN-50-2 (Salman et al., 2020) with $s_{\mathbf{y}} = 8$. For $s_{\mathbf{x}} = 0$, the setup equals the GAS setup and synthesizes an image unrelated to the input. As the scale increases, the image gets closer to the original.

## B.3    Additional classifiers for adversarial attacks using GAT

To verify the efficacy of ScoreAG and demonstrate its applicability across various architectures, we evaluate the accuracy of GAT on four more pretrained classifiers via `PyTorch Hub`[2] for the datasets CIFAR10 and CIFAR100 using the same hyperparameters, i.e., scale parameters, as for the WRN-28-10 classifier. We show the adversarial accuracy in Tab. 8, including selected baselines. As we can observe, ScoreAG successfully generates adversarial attacks on various classifiers, reaching accuracies close to 0%. This demonstrates the flexibility of ScoreAG and applicability to arbitrary pre-trained classifiers.

---

[2]https://github.com/chenyaofo/pytorch-cifar-models

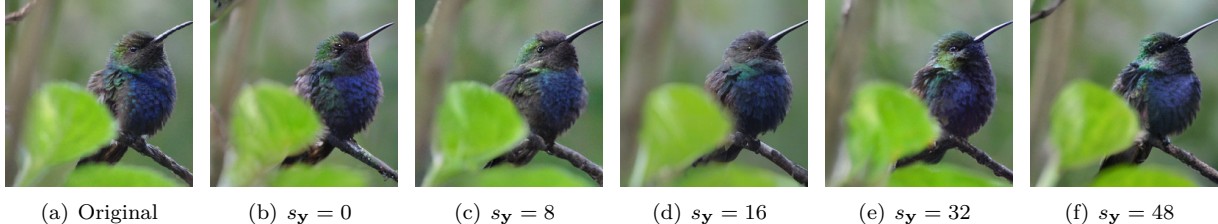

| (a) Original | (b) $s_{\mathbf{y}} = 0$ | (c) $s_{\mathbf{y}} = 8$ | (d) $s_{\mathbf{y}} = 16$ | (e) $s_{\mathbf{y}} = 32$ | (f) $s_{\mathbf{y}} = 48$ |

Figure 7: Effect of the scale parameter $s_{\mathbf{y}}$. The images display adversarial images generated by ScoreAG-GAT across different scales $s_{\mathbf{y}}$ on a robust WRN-50-2 (Salman et al., 2020) with $s_{\mathbf{x}} = 0.25$. For $s_{\mathbf{y}} = 0$, the setup equals the GAP setup and synthesizes an image without adversarial perturbations. As the scale increases, the adversarial content strengthens, causing the images to diverge further from the original.

Table 8: Adversarial accuracy of ScoreAG for various classifiers on the datasets CIFAR10 and CIFAR100.

| | ResNet-20 | | ResNet-56 | | VGG-19 | | RepVGG-A2 | |
| Dataset | CIFAR10 | CIFAR100 | CIFAR10 | CIFAR100 | CIFAR10 | CIFAR100 | CIFAR10 | CIFAR100 |
| --- | --- | --- | --- | --- | --- | --- | --- | --- |
| **$\ell_\infty$ restricted** | | | | | | | | |
| FGSM (Goodfellow et al., 2014) | 14.95 | 4.97 | 34.33 | 8.09 | 29.96 | 20.87 | 51.37 | 10.20 |
| DI-FGSM (Xie et al., 2019) | **0.00** | 0.01 | 0.12 | **0.00** | 1.06 | 2.17 | 1.55 | 2.17 |
| SI-NI-FGSM (Lin et al., 2019) | 0.51 | 0.14 | 2.12 | 0.55 | 11.89 | 4.74 | 4.27 | 4.74 |
| APGD (Croce & Hein, 2020b) | **0.00** | 0.01 | 0.01 | 0.01 | 0.14 | 0.77 | 0.06 | 0.77 |
| APGDT (Croce & Hein, 2020b) | **0.00** | **0.00** | **0.00** | **0.00** | **0.00** | **0.00** | **0.00** | **0.00** |
| Square (Andriushchenko et al., 2020) | **0.00** | **0.00** | **0.00** | **0.00** | 0.51 | 0.76 | 0.42 | 0.11 |
| FAB (Croce & Hein, 2020a) | 0.29 | 0.36 | 0.31 | 0.35 | 4.79 | 2.67 | 1.58 | 0.18 |
| **$\ell_2$ restricted** | | | | | | | | |
| APGD (Croce & Hein, 2020b) | **0.00** | 0.01 | 0.01 | 0.01 | 0.14 | 0.77 | 0.06 | 0.02 |
| APGDT (Croce & Hein, 2020b) | **0.00** | **0.00** | **0.00** | **0.00** | **0.00** | **0.00** | **0.00** | **0.00** |
| Square (Andriushchenko et al., 2020) | **0.00** | **0.00** | **0.00** | **0.00** | 0.51 | 0.76 | 0.42 | 0.11 |
| FAB (Croce & Hein, 2020a) | 0.25 | 0.40 | 0.30 | 0.31 | 4.79 | 2.67 | 1.53 | 0.14 |
| **$\ell_0$ restricted** | | | | | | | | |
| OnePixel (Su et al., 2019) | 76.39 | 42.28 | 81.00 | 44.60 | 74.49 | 45.47 | 82.79 | 56.17 |
| **Unrestricted** | | | | | | | | |
| CAA (Hsiung et al., 2023) | 25.10 | 4.05 | 36.75 | 5.16 | 33.75 | 10.20 | 43.16 | 7.43 |
| PPGD (Laidlaw et al., 2020) | 50.93 | 29.76 | 43.88 | 32.52 | 10.95 | 23.33 | 33.14 | 33.53 |
| LPA (Laidlaw et al., 2020) | **0.00** | **0.00** | **0.00** | **0.00** | 0.02 | 0.30 | 0.01 | **0.00** |
| ScoreAG (Ours) | **0.00** | **0.00** | **0.00** | **0.00** | 0.02 | 0.16 | **0.00** | **0.00** |

Additionally, we evaluate ScoreAG on the high-resolution ImageNet-Compatible[3] dataset, a commonly used subset of ImageNet. We selected two robust classifiers as most attacks achieved 0% accuracy on standard classifiers. More specifically, we selected the RaWideResNet-101-2 by Peng et al. (2023) and WideResNet-50-2 by Salman et al. (2020). We show the results for ScoreAG and selected baselines, including Adversarial Content Attack (Chen et al., 2023c) (ACA), in Tab. 9. The restricted baselines have a perturbation distance of $\ell_p = 4/255$. As we can observe, ScoreAG again achieves competitive performance, i.e., best and second-

Table 9: Adversarial accuracy of ScoreAG for robust classifiers on the ImageNet-Compatible dataset.

| | Salman et al. (2020) | Peng et al. (2023) |
| --- | --- | --- |
| **$\ell_\infty$ restricted** | | |
| FGSM (Goodfellow et al., 2014) | 58.8 | 66.5 |
| DI-FGSM (Xie et al., 2019) | 57.6 | 66.8 |
| SI-NI-FGSM (Lin et al., 2019) | 74.5 | 80.2 |
| APGD (Croce & Hein, 2020b) | 52.2 | 62.3 |
| APGDT (Croce & Hein, 2020b) | 46.5 | 59.1 |
| **$\ell_0$ restricted** | | |
| OnePixel (Su et al., 2019) | 85.3 | 88.2 |
| **Unrestricted** | | |
| CAA (Hsiung et al., 2023) | 10.4 | 11.9 |
| PPGD (Laidlaw et al., 2020) | 5.9 | 18.5 |
| LPA (Laidlaw et al., 2020) | **1.6** | 8.8 |
| DiffAttack (Chen et al., 2023a) | 6.0 | 8.5 |
| ACA (Chen et al., 2023c) | 4.6 | 6.5 |
| ScoreAG (Ours) | 2.5 | **4.1** |

---

[3]https://github.com/cleverhans-lab/cleverhans/tree/master/cleverhans_v3.1.0/examples/nips17_adversarial_competition/dataset.

best accuracies. We show some examples of the unrestricted attacks in Fig. 8. As expected, ScoreAG preserves the semantics of the images and displays a high degree of realism. Surprisingly, the other diffusion-based attacks, DiffAttack and ACA, display more noticeable differences. ACA, in particular, has made major changes to the image.

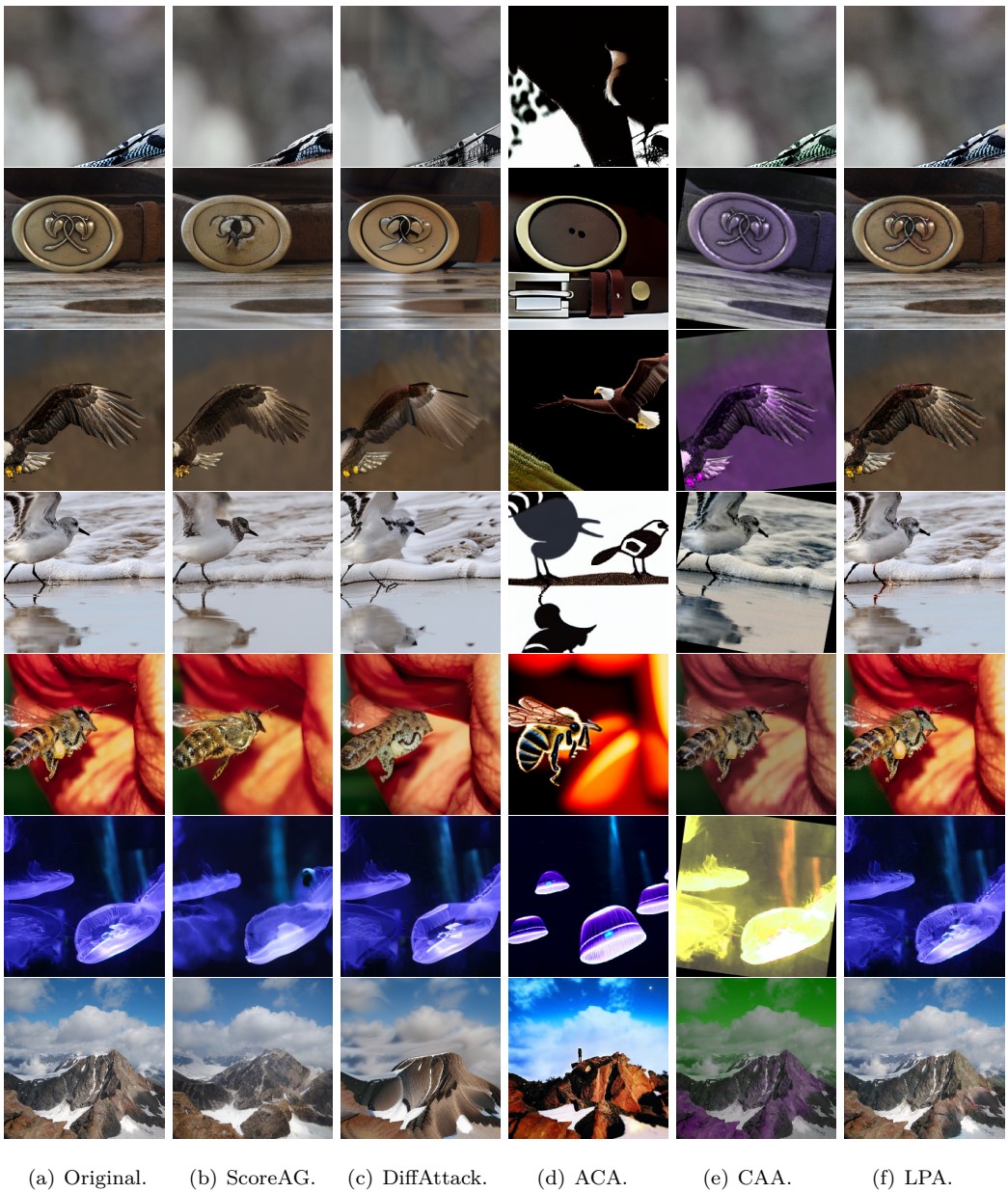

(a) Original.     (b) ScoreAG.     (c) DiffAttack.     (d) ACA.     (e) CAA.     (f) LPA.

Figure 8: Adversarial examples on the ImageNet-Compatible dataset of various classes for different unrestricted attacks.

## B.4 Purification of Common Corruptions

In addition to the purification of adversarial attacks, we test the applicability of ScoreAG (GAP) on common corruptions (Hendrycks & Dietterich, 2019). We show the robust accuracy of standard and robust classifiers before and after purification using DiffPure and GAP in Tab. 10. Our results show that ScoreAG consistently increases the robust accuracy over the base model. In 5/7 settings, ScoreAG achieves a better accuracy than

Table 10: CIFAR10 robust accuracy of different adversarial training and purification methods for common corruptions on CIFAR10. If multiple threat models exist, we denote results as $\ell_\infty/\ell_2$. Best scores are bolds.

| Model | Base | DiffPure (Nie et al., 2022) | ScoreAG-GAP (Ours) | Architecture |
|---|---|---|---|---|
| Standard | 75.56±0.41 | 81.85±0.59 | **83.47±1.25** | WRN-28-10 |
| **Adversarial Training** | | | | |
| (Cui et al., 2023) | 81.90 | **82.76** | 82.32 | WRN-28-10 |
| (Wang et al., 2023) | 81.38 / 87.96 | **81.98** / 86.40 | 81.58 / **88.74** | WRN-28-10 |
| (Wang et al., 2023) | 83.90 / 89.24 | 84.16 / 87.14 | **84.30 / 89.86** | WRN-70-16 |
| (Peng et al., 2023) | 83.32 | 83.76 | **83.94** | RaWRN-70-16 |

DiffPure. Surprisingly, purifying the standard model makes it competitive with adversarially trained models, implying that purification does only benefit little when combined with a robust classifier.

## B.5 Large Perturbation Norms for restricted adversarial attacks

In Fig. 9, we show adversarial examples of different attacks for the image in Fig. 1. We use the same distances ScoreAG achieves.

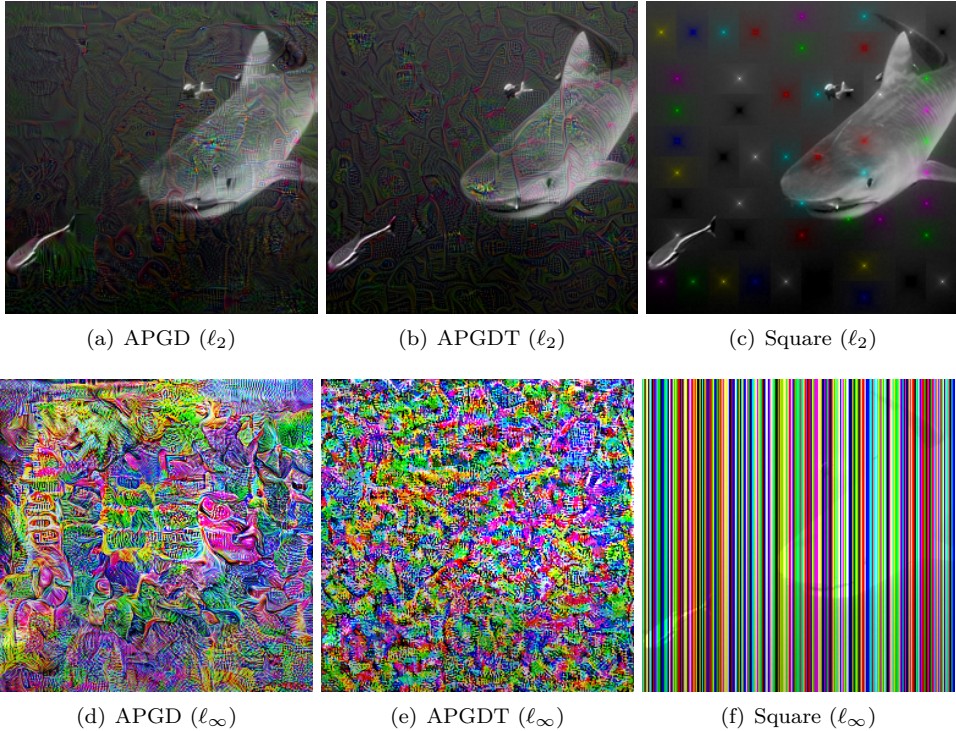

(a) APGD ($\ell_2$)    (b) APGDT ($\ell_2$)    (c) Square ($\ell_2$)

(d) APGD ($\ell_\infty$)    (e) APGDT ($\ell_\infty$)    (f) Square ($\ell_\infty$)

Figure 9: Different adversarial attacks for the example in Fig. 1. The $\ell_\infty$ and $\ell_2$ distances are 188/255 and 18.47, respectively. All methods display major changes in the images compared to the original.

## B.6 Generative Adversarial Synthesis

In Fig. 10, we provide additional examples of the GAS task. The images are synthetic adversarial samples of the ImageNet class "indigo bunting". While all images are classified wrongly, most of them contain the right core-semantics and display a high degree of realism.

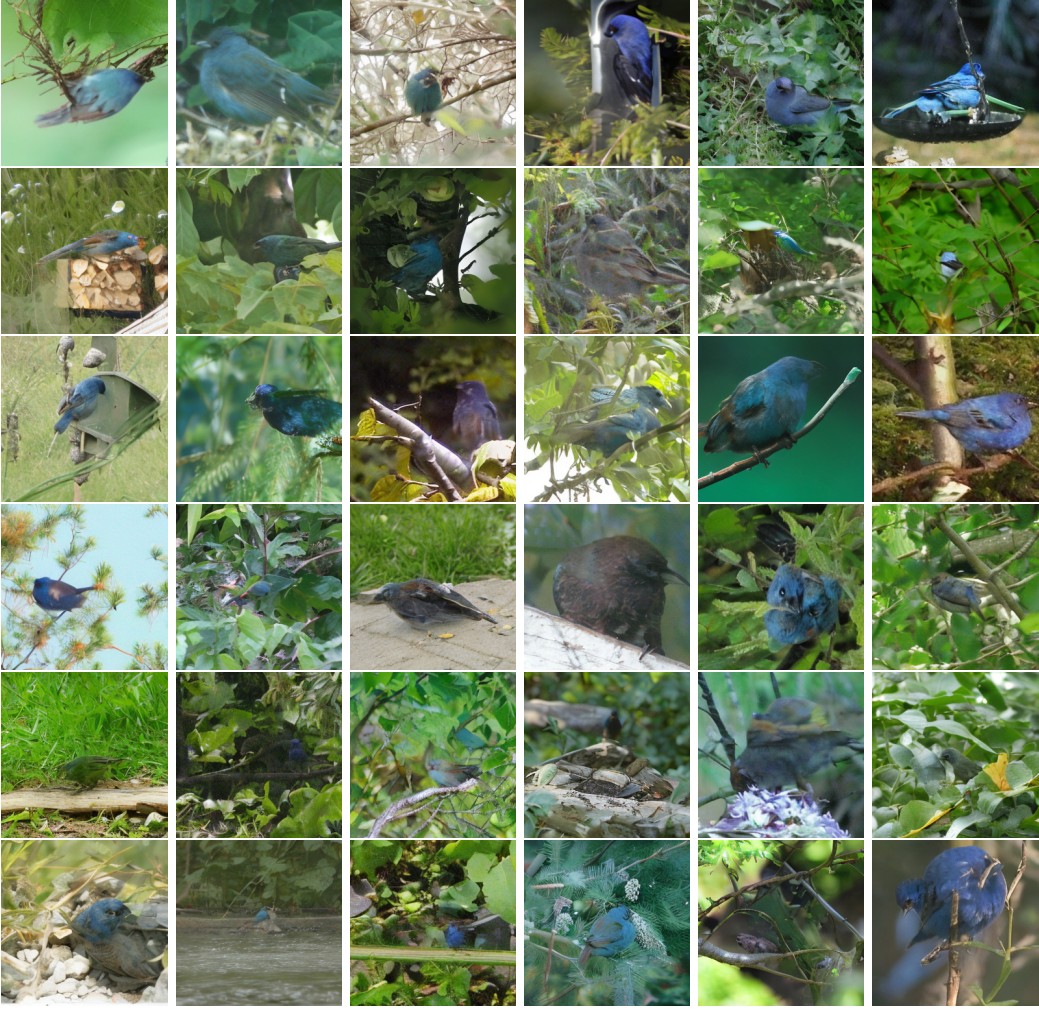

Figure 10: Selected synthetic adversarial examples on ImageNet for the class "indigo bunting". All images display a high degree of realism and are classified wrongly into various classes.

### B.7  Generative Adversarial Transformation

In Fig. 11, we show additional examples of the GAT task. All original images are classified correctly into the ImageNet classes "golden retriever", "spider monkey", "football helmet", "jack-o'-lantern", "pickup truck", and "broccoli", while the adversarial images are classified as "cocker spaniel", "gibbon", "crash helmet", "barrel", "convertible", and "custard apple", respectively. While all adversarial images display subtle differences they do not alter the core semantics of the images and are not captured by common $\ell_p$-norms.

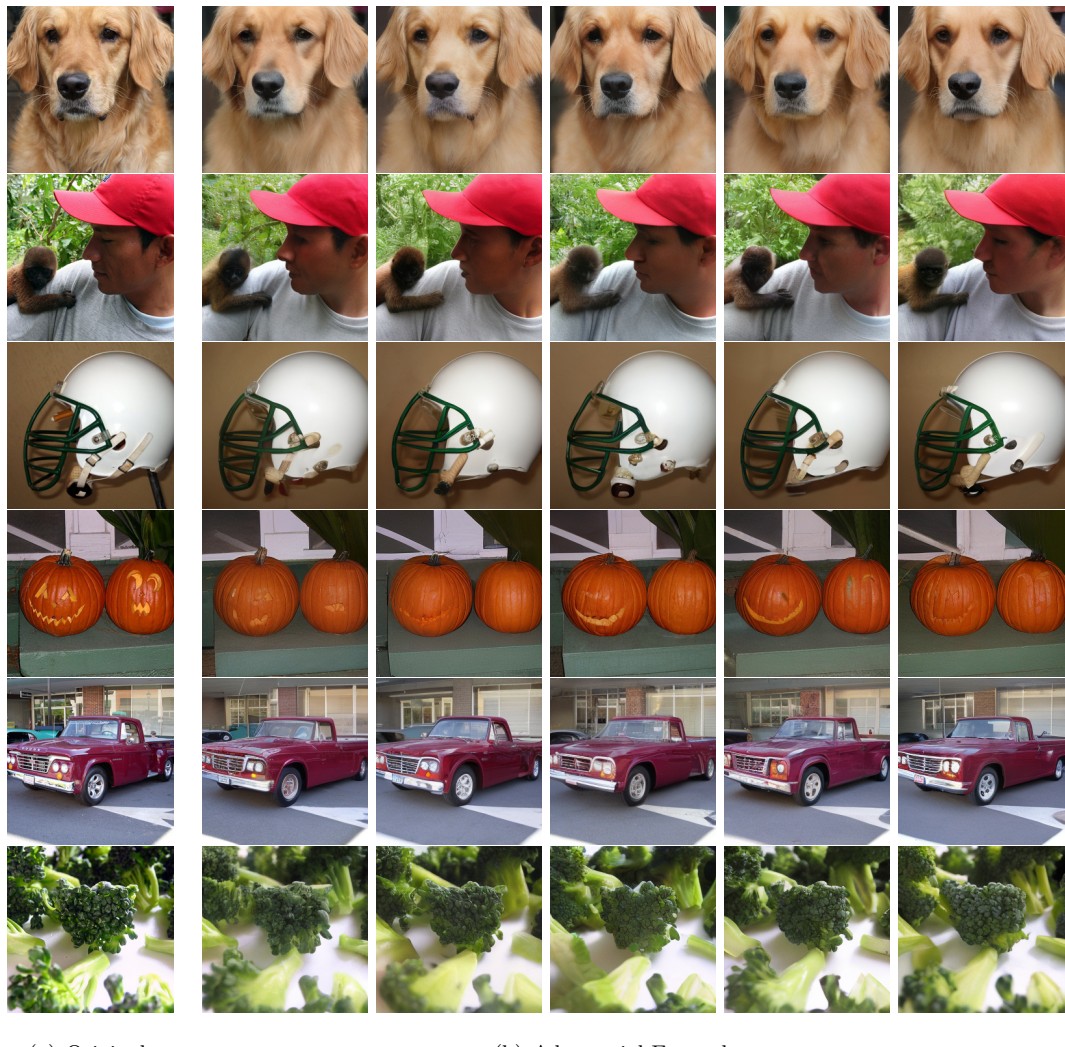

(a) Original.  (b) Adversarial Examples.

Figure 11: Selected transformed adversarial examples on ImageNet. While the adversarial examples are classified wrongly, the original images are classified correctly. All images maintain the semantics while being outside of common perturbation norms.

### B.8  Runtime comparison of the attacks.

All experiments were conducted on A100s. In Tab. 11, we report the runtimes in seconds of various methods. The numbers display the average time to generate one adversarial example on the ImageNet-Compatible dataset. Note that sampling an image without guidance using the same generative model as ScoreAG takes 15.00 seconds. The difference stems from the additional overhead induced by the gradient computations.

Table 11: Average runtimes in seconds of the different attacks on an A100 to generate one adversarial images for the ImageNet-Compatible dataset.

| FGSM | DIFGSM | SINIFGSM | Square | FAB | APGD | APGDT | OnePixel | LPA | PPGD | DiffAttack | ACA | ScoreAG |
|------|--------|----------|--------|-----|------|-------|----------|-----|------|------------|-----|---------|
| 0.45 | 0.18 | 0.68 | 36.04 | 74.39 | 0.40 | 1.86 | 0.64 | 0.96 | 0.50 | 19.14 | 188.80 | 79.34 |

