# OpenReview forum: "Assessing Robustness via Score-Based Adversarial Image Generation"
_TMLR — Accepted by TMLR_

### Review · Reviewer_2bFN · 2024-08-23

**Summary Of Contributions:**

Summary:
This paper studies adversarial learning using the diffusion process. Specifically, existing adversarial generation would change the semantic information of the image data, hence, the authors propose score-based diffusion to conduct adversarial generation, which can largely preserve the original semantic feature. The score-based diffusion process requires a task-specific condition to guide the learning process. There are three different tasks are considered, namely generative adversarial synthesis, generative adversarial transformation, and generative adversarial purification. Through quantitative experimental analysis, the authors show that the proposed diffusion method can achieve improved performance compared to existing baseline method.

**Audience:**

Yes

**Broader Impact Concerns:**

The proposed method is interesting, and could potentially helpful for the adversarial field.

**Claims And Evidence:**

Yes

**Requested Changes:**

Please see the weakness part.

**Strengths And Weaknesses:**

Strengths:
- This paper studies very interesting topic and the diffusion process can effectively improve the quality of adversarial attack, compared to existing methods, the adversary is semantically similar to the natural examples.
- This paper is well-organized and well-written. The content of this paper is sufficient.
- The performance improvement is promising.


Weaknesses:
- Since the proposed method employs diffusion strategy, the computational burden could be largely increased. However, there is no detailed analysis about the computational efficiency.
- Lack of intuitive qualitative analysis to understand the proposed method better. Although the effectiveness of the proposed method has been largely benefited from the diffusion process, however, why such a process can help the learning result is still unknown. It is suggested to conduct further visualizations to provide better intuition.

---

### Review · Reviewer_sFqw · 2024-09-07

**Summary Of Contributions:**

The paper introduces a new framework to generate adversarial examples. The authors use score-based diffusion models to generate unrestricted (in the sense of being out of an l_p norm bound) adversarial images. The resulting images preserve the semantics of the true labels. The framework can accomplish three tasks: a) generation of new adversarial images from scratch b) transformation of the existing images into adversarial ones, c) image purification (for training) to enhance classifiers’ robustness against adversarial attacks.

**Audience:**

Yes

**Claims And Evidence:**

Yes

**Requested Changes:**

Please make changes based on what I pointed out above, in the Weaknesses part.

**Strengths And Weaknesses:**

**Strengths**:

-- The paper is mostly well written and structured and it’s easy to follow, except for the background section.

-- The experimental evaluations are mostly comprehensive -- they compare against several baselines and include multiple datasets.

-- Adversarial examples can exist outside of the conventional l_p norms and diffusion generative models seem a natural choice here.

**Weaknesses**:

-- While the Background Section is crucial to fully understand ScoreAG, it’s not clear at all. The definitions, notations and equations are too concise and their connections to ScoreAG’s components are not explained. I suggest adding more explanation to the definitions in section 2 and making it clear how they would constitute ScoreAG. For example, in the “diffusion guidance” paragraph, “… denotes the gradient of a classifier and c the guidance condition” needs more details.

-- I suggest the authors add a problem formulation section and explain some of the terms and notations there. This makes understanding Section 3 much easier. For example, the term ‘task’ is used frequently but not defined formally. Also, the "adversary" paragraph could be moved to a potential problem formulation section.

-- In the experimental setup, mention that the ± indicates the 95th CI over 4 runs, across all the experiments in the main text.

-- In Table1, why is no row bolded under LPIPS? Furthermore, some numbers corresponding to CAA and DiffAttack are significantly higher than the others. I wonder why this is the case. Maybe if you include them systematically in Figure 3, we can make a better sense of it.

-- The order of the figures is not preserved in the text. Fig 3 is cited after Fig 4.

-- In the caption of Figure 3 (also in the text), what are the example values of “common perturbation norms”?

-- In Table 2, why only the CIs of the last row are reported? Table 5 does not report any CI at all?

-- In the “Evaluating Generative Adversarial Transformation” paragraph, you attribute the lower attack success rate of the other diffusion-based methods to the fact that they only leverage the last few iterations of the denoising diffusion process. Can you elaborate on the intuition? Also are there any experiments to support your claim?

---

### Review · Reviewer_mc5Z · 2024-09-30

**Summary Of Contributions:**

The paper proposes to use score-based generative models to tackle three tasks in the domain of adversarial attacks: synthesis, transformation, and purification. The paper proposes to do these three tasks in an (IMO) elegantly unified manner: by modifying the score function that is followed during the denoising process. Each of these three tasks is thus reduced to adding a term in the score function, considering the information that is provided for each of task. The individual tasks are then evaluated on standard (albeit small, by today's standard in computer vision in general) datasets. The proposed approaches work reasonably well, and the paper does a commendable job of measuring how conceptually-changing the synthesis and transformations are.

**Audience:**

Yes

**Broader Impact Concerns:**

Not any that I can think of.

**Claims And Evidence:**

Yes

**Requested Changes:**

Please refer to "Weaknesses that affect my rating" for changes I request. Some of them may be provided in the rebuttal, while others should be incorporated in the manuscript. For the other section "Weaknesses that don't affect my rating, ..." please change as most as possible.

Some other minor complaints:
- Another minor complaint w.r.t. notation: can the paper use \eqref?
- Extremely minor complaint about notation: the scaling parameters in the equation relating to the score function (say $s_x$) would look better with the corresponding boldface: $s_{\textbf{x}}$
- Overall, I'd encourage not to say "FID score", since it's a distance, so lower is probably better, while a score sounds like higher is better
- I think Table 1 would be more readable with single-digit precision for the numbers. Right now it looks rather cluttered
- I think Fig. 4 is referenced before Fig. 3. Maybe switch the order?
- The table in P8 is labeled as a figure (see caption)
- In section 4.3 I'd use the word "distinguishable" instead of "differentiable"
- P9, last line: clerical error: ").In"
- P10 clerical error: "inpainting.Unlike"
- P10: Kang et al. (2023) *have*

**Strengths And Weaknesses:**

Strengths:
- The paper proposes and validates a useful way to leverage current generation models to study the adversarial robustness of classifiers. I think this is an important point, since the usual setup of adversarial robustness has become, in my humble opinion, obsolete.
- Overall, the paper has commendable presentation: both in terms of language, quality of figures, and organization.
- The high performance of the purification method, combined with the fact that by construction it is applied on all images, suggests it is most likely also a good detector no? If you purify and the original label differs from the new label then that instance could be predicted to be adversarial?
- The purification method is even tested on common corruptions

---
Weaknesses that affect my rating:
- Could the authors elaborate on the intuition behind the purification method? By the looks of the equation in 3.3, it seems like the denoising process is conducted essentially with the hopes that the model will recover the true class of the image, without further information about the image. If my intuition is correct, does this mean that the model is kind of expected to map the image back to the manifold of unperturbed images by itself? Or is there something else I'm missing?
- For evaluation of the purification method, I understand the argument for performing the method on top of black-box-generated adversarial examples. However, do the authors know how well/bad does the proposed purification method work on white-box attacks?
- For the evaluation metrics, I'm unsure about the usage of FID. That distance is for distributions, while in this case we would most likely intend for something at the instance level. Could the authors elaborate more on their rationale for choosing FID?
- Section 4.2 mentions qualitative experiments on ImageNet. Is there any specific reason why quantatitve results are not reported?
- For the Related Work section on Adversarial Attacks, I think the paper should explicitly mention to which line of work the proposed transformation method belongs to (i.e. white box, correct? since you need the model "f" to compute the (modified) score function)
- I found that the computational cost of running the attack is reported on the appendix. I think the main paper should point out that the appendix reports that information
- Table 9 reports that ScoreAG requires about 80 seconds to run. Could the authors disagregate how much of that time is the "regular" denoising process itself vs. how much is actually added by the proposed terms in the score function (i.e. the overhead). I think that's a more useful piece of information, since most likely the coming months/years will see dramatic improvement in the efficiency of the denoising process itself.

-----
Weaknesses that *don't* affect my rating, but should be addressed to improve the paper:
- Fig. 1 is a case of (in the paper's vocabulary) Generative Adversarial Transformation (Sec. 3.2), correct? If so, I think explicitly stating this fact would be informative for the figure's caption (to convey that it is one out of three things that the paper proposes).
- All three subsections of section 3 are correct (IMO). However, in the interest of  clarity, I think the paper would benefit from explicitly stating a clearer explanation of how all synthesis, transformation, and purification are performed in practice by using either (1) a formal definition, or (2) an algorithm. I would advocate for the algorithm, since I don't see the paper reporting that it will provide an open source codebase.
- P5L3 explicitly states "<unrestricted> attack". I think I understand the intention, but I would not really call this unrestricted, but rather that the restriction is much more sophisticated (and larger) than the \ell-p restriction.
- The second-to-last line in Sec. 3.3 says "they also". What subject is that? ScoreAG?
- The first sentence in Sec. 4 states "assess the capability of ScoreAG in generating adversarial examples." I don't think that statement accurately reflects the rest of the section: the evaluation is not just about capability to generate adversarial examples, is it?
- Do the authors have thoughts on if (and how) classifier-free guidance could be used here?

---

### Decision · Action_Editor_AXh7 · 2024-11-11

**Recommendation:** Accept as is

**Comment:**

This paper proposes a new "semantically-preserving" adversarial attack to evaluate the robustness of vision models beyond lp perturbations. This is an area of interest of the community. All reviewers agree this is a solid work based on an interesting idea and with a clear presentation.

**Audience:**

This paper proposes a new technique to leverage diffusion models to evaluate the robustness of vision models to semantically-preserving transformations of an image. This is an area of clear interest to the TMLR audience as highlighted by all reviewers.

**Claims And Evidence:**

All reviewers agree that this paper provides accurate, convincing and clear evidence to support its claims. It contains both quantitative and qualitative evaluations of the proposed method under different settings, and even the more subjective property of "semantic-preservation" of the attack is tested using a human study.